# Fine and coarse dust radiative impact during an intense Saharan dust outbreak over the Iberian Peninsula - long-wave and net direct radiative effect

- María Ángeles López-Cayuela<sup>1</sup>, Carmen Córdoba-Jabonero<sup>1\*</sup>, Michaël Sicard<sup>2,#</sup>, Jesús Abril-Gago<sup>3,4</sup>, Vanda Salgueiro<sup>5</sup>, Adolfo Comerón<sup>2</sup>, María José Granados-Muñoz<sup>3,4</sup>, Maria João Costa<sup>5</sup>, Constantino Muñoz-Porcar<sup>2</sup>, Juan Antonio Bravo-Aranda<sup>3,4</sup>, Daniele Bortoli<sup>5</sup>, Alejandro Rodríguez-Gómez<sup>2</sup>, Lucas Alados-Arboledas<sup>3,4</sup> and Juan Luis Guerrero-Rascado<sup>3,4</sup>
- ¹Instituto Nacional de Técnica Aeroespacial (INTA), Atmospheric Research and Instrumentation Branch, Torrejón de Ardoz, 28850-Madrid, Spain
  - <sup>2</sup>CommSensLab, Dept. of Signal Theory and Communications, Universitat Politècnica de Catalunya (UPC), 08034-Barcelona, Spain
  - <sup>3</sup>Andalusian Institute for Earth System Research (IISTA-CEAMA), 18006-Granada, Spain
- <sup>4</sup>Department of Applied Physics, University of Granada (UGR), 18071-Granada, Spain
  - <sup>5</sup>University of Évora, School of Sciences and Technology, Department of Physics, Center for sci-tech Research in EArth sysTem and Energy (CREATE), 7004-516 Évora, Portugal
  - \*Now at: Laboratoire de l'Atmosphère et des Cyclones (LACy), Université de La Réunion, Saint Denis, France
  - Correspondence to: Carmen Córdoba-Jabonero (cordobajc@inta.es)

- **Abstract.** The dust direct radiative effect (DRE) in long-wave (DRE<sub>LW</sub>), and net effect (DRE<sub>NET</sub>), is analysed during an intense and long-lasting Saharan dust intrusion over the Iberian Peninsula, complementing the study on the short-wave DRE (DRE<sub>SW</sub>) (López-Cayuela et al., 2025). In LW, a warming effect at both bottom-of-atmosphere (BOA) and the top-of-atmosphere (TOA) levels is induced by the fine (Df) and coarse (Dc) dust particles (Dc dominant). The DRE<sub>LW</sub>-to-DRE<sub>SW</sub> ratio for Df ranged 4-
- 8% at BOA (1-4% at TOA), and for Dc it was rather higher (39-54% at BOA and 20-50% at TOA). DRE<sub>NET</sub> was consistently negative (net cooling) at both levels, and hence the atmospheric DRE<sub>NET</sub> was positive (net warming). The Df contribution to DRE<sub>NET</sub> was 12% (LW) and 30% (SW). The SW aerosol heating rate (AHR) peaked at higher altitudes, inducing warming within the dust layer, than LW AHR (weaker cooling). Consequently, a net warming inside the dust layer was found, with potential cooling below and above. While SW dominates the net atmospheric warming, LW cooling partially mitigates it.
- DRE<sub>LW</sub> (and DRE<sub>NET</sub>) is underestimated (overestimated) by using the dust-mode separation approach when fine radii are lesser (greater) than a particular threshold (e.g., 0.1 μm), revealing the particle size impact in DRE<sub>LW</sub>. The dust-induced net effect is primarily driven by SW and modulated by LW. The classical (no separation) approach overestimates DRE<sub>NET</sub>, with mean relative differences of -5%/-9% at BOA/TOA. Moreover, under moderate-to-high dust, separating Df and Dc contributions yields a weaker (stronger) net cooling at BOA (TOA).

#### 1 Introduction

- The latest report about airborne dust from the World Meteorological Organization (WMO Bulletin, 2023) reveals that the global surface dust concentration has seen a slight increase in 2022 compared to 2021. This is attributed to increased emissions from several dust-active sources, such as west-central Africa. Among the most affected regions which receive this dust influx, which is much greater than the climatological mean, the Iberian Peninsula (IP) is prominently featured. Particularly, the anomaly of the annual mean surface dust concentration in 2022 (relative to the 1981-2010 mean) shows increased values of 5-
- 20 µg m<sup>-3</sup>. Those results agree with several different studies pointing out that since the pre-industrial Era there has been a 46% increase in the mass of dust lifted into the atmosphere in these North African regions (Kok et al., 2023). Notably, not only there is a rising frequency of Saharan dust episodes in the IP, as compared to long-term historical data (Sousa et al., 2019), but also a growing number of studies reporting extreme and highly intense episodes (e.g., Guerrero-Rascado et al., 2009; Preißler et al., 2011; Cazorla et al., 2017; Córdoba-Jabonero et al., 2019; Fernández et al., 2019; López-Cayuela et al., 2023;
- Papanilokaou et al., 2024). Those results gain significance since desert dust aerosols affect Earth's energy balance. Thus, variations in the atmospheric dust loading may induce substantial changes in the radiative forcing of the climate system (Mahowald et al., 2010).
  - The aerosol radiative effect in the short-wave (SW) spectral range related to desert dust intrusions over the IP has been widely investigated during the last years (e.g. Cachorro et al., 2008; Obregón et al., 2015; Sicard et al., 2016; Valenzuela et al., 2017;
- Granados-Muñoz et al., 2019; Córdoba-Jabonero et al., 2021; Bazo et al., 2023, López-Cayuela et al., 2025). However, part of the literature often overlooked the aerosol radiative effects on the long-wave (LW) spectral range. This omission was primarily attributed to the intricate challenges associated with precisely quantifying the optical characteristics within this spectral domain (Roger et al., 2006; Mallet et al., 2008; Sicard et al., 2012). Moreover, the radiative forcing (RF) attributed to most aerosol categories (in particular, fine particles like pollution and smoke), is generally less pronounced in the LW range in contrast to
- their effects in the short-wave (SW) range. But an exception arises with large and light-scattering particles (like mineral dust), which have been shown to possess a significant RF effect in the LW domain (e.g., Fouquart et al., 1987; di Sarra et al., 2011; Sicard et al., 2014a, 2022), highlighting again its climatic importance.
  - The present paper focuses on the assessment of the direct radiative effect (DRE) of dust particles in the LW range as well as their net effect. The event in study is an exceptionally intense and long-lasting Saharan dust event that crossed the IP from 25
- March to 7 April 2021. The investigation was carried out using data obtained from five Iberian lidar stations: El Arenosillo/Huelva (ARN), Granada (GRA), Torrejón/Madrid (TRJ), and Barcelona (BCN) in Spain, and Évora (EVO) in Portugal. Moreover, the use of lidar measurements together with the POLIPHON method (Polarisation Lidar photometer Networking method; Mamouri and Ansmann, 2014, 2017; Ansmann et al., 2019), is an added value to derive the vertical distribution of dust, which can be split in their fine and coarse contributions. The optical and microphysical properties were
- reported in López-Cayuela et al. (2023), and the DRE effect in the SW range can be found in López-Cayuela et al. (2025). Both studies specify the locations of the stations and the days on which dust intrusion was observed; for the reader's convenience, that information can also be found in Table S1 in the Supplementary Material (SM).
  - This work is an added value in this field. Although several works investigated the LW radiative effects associated with desert dust outbreaks over the Mediterranean basin (e.g., di Sarra et al., 2011; Perrone et al., 2012; Antón et al., 2014; Bazo et al.,
- 2023), only a few studies have addressed the separation of both components (e.g., Sicard et al., 2014b, 2022). Two main conclusions highlight from these studies (i) the quasi-linearity of LW RF at the bottom-of-atmosphere (BOA) and top-of-atmosphere (TOA) with the Aerosol Optical Depth (AOD), and (ii) the high dependency of LW RF on the coarse-mode dust. Thus, the aim of the present study is to investigate whether this quasi-linearity holds for high AOD values (> 0.50), in addition to distinguish the contribution of the fine dust (Df) and coarse dust (De) components to the DRE in the LW range. Indeed, this
- study introduces the novelty of simulating the LW dust DRE using two different approaches, as performed in López-Cayuela et al. (2025) for the SW range: (i) by simulating the contribution of Df and Dc components separately, and then estimating the total dust DRE as their sum (as DD = Df + De), and (ii) directly simulating DRE for the total dust component as a whole.

The paper is organized as follows. The radiative transfer model and the parametrizations used in terms of the LW range are described in Section 2. The results and discussion are shown in Section 3. Finally, the main conclusions of this study are found in Section 4.

#### 2. Methodology

#### 2.1 Radiative transfer model: GAME. The MIE and LW modules

The GAME (Global Atmospheric Model; Dubuisson et al., 1996, 2004) has been used in increasing studies because of its significant advantage, i.e. the ability to fully represent the aerosol scattering and absorption in the LW region. Moreover, the model's moderate spectral resolution accounts for the spectral variations in aerosol properties, particularly in the infrared window. An extended description of the LW module of GAME can be found in Sicard et al. (2014a).

GAME calculates spectrally integrated upward and downward radiative fluxes in 40 plane and homogeneous layers from 0 to 100 km. Regarding the spectral limits, GAME employs 200 to 2500 cm<sup>-1</sup> (i.e. wavelength: 4.0–50.0 μm) with a fixed resolution of 20 cm<sup>-1</sup> (115 points). Moreover, this radiative transfer model considers thermal emission, absorption and scattering as well as their interplay employing the discrete ordinates method (DISORT, Stamnes et al., 1988). In the framework of GAME, an explicit account is taken for the absorption of gases, including H<sub>2</sub>O, CO<sub>2</sub>, O<sub>3</sub>, N<sub>2</sub>O, CO, CH<sub>4</sub>, and N<sub>2</sub>, using the correlated k-distribution as proposed by Lacis and Oinas (1991). Detailed insights into the computation of gas transmission functions can be found in Dubuisson et al. (2004) and Sicard et al. (2014b). The parameterization of gas absorption is based on pressure, temperature, and relative humidity profiles. Notably, these profiles are sourced from the Global Data Assimilation System (GDAS), provided by the National Oceanic and Atmospheric Administration (NOAA, last access: 28 March 2025).

The land surface temperature (LST) is a variable needed in the LW module. In this work, LST is provided by the Copernicus Land Service (https://land.copernicus.eu/global/products/lst, last access: 28 March 2025). Particularly, the hourly LST V2 dataset is used. Moreover, the Earth's surface is assumed Lambertian, with a constant surface albedo (SA) of 0.017 in the LW spectral range. This value was determined by Sicard et al. (2014a) in Barcelona, based on the Clouds and Earth's Radiant Energy System (CERES) measurements in the spectral range of 8.1–11.8 µm, and averaged over the spring and summer seasons during five years. This same value is used at the five stations of this study, in the basis of the work of Zhou et al. (2013), which showed that the LW surface albedo remains relatively stable across the European continent.

Information on the aerosol shape, refractive index, size distribution, and density is required for an accurate calculation of their radiative properties. Yang et al. (2007) demonstrated that the non-sphericity effect of dust particles is negligible at thermal infrared wavelengths. Therefore, it is reasonable to assume that mineral dust is 'spherical' in the LW range, and, hence, a Mie code can be applied for analysis. The spectral refractive index (both real and imaginary components) is identical to that reported in Sicard et al. (2014a) and was derived from measurements of long-range transported mineral dust collected in western Germany (Volz, 1983). The data that present the refractive index as a function of wavelength was obtained from Krekov (1993). The spectral variation of both the real and imaginary parts of the refractive index is illustrated in Figure 1 of Sicard et al. (2014a). Moreover, the geometric median radius ( $r_g$ ), and its standard deviation ( $σ_g$ ), of the lognormal distribution are also needed in the Mie code. Those parameters are obtained for both the coarse and fine modes using column-integrated AERONET (Aerosol Robotic NETwork; <a href="http://aeronet.gsfc.nasa.gov">http://aeronet.gsfc.nasa.gov</a>) Version 3 Level 2.0 data inversion products (last access: 28 March 2025). AERONET provides the volume median radius ( $r_v$ ) and its corresponding standard deviation ( $σ_v$ ); hence, the following expressions were applied to determine both  $r_g$  and  $σ_g$ :

$$r_g = r_v e^{-3\left(\ln\sigma_g\right)^2},\tag{1}$$

being  $\sigma_g = \sigma_v$ . These data were hourly averaged (and interpolated if missing). The column-integrated number concentration (N) is also derived. The AERONET column-integrated volume concentration  $(v_c)$ , together with the  $r_v$  and  $\sigma_v$ , is used to calculate N as follows:

$$125 \quad N = 3 \frac{v_c}{4\pi r_s^3 \sqrt{2\pi} log \sigma_v}. \tag{2}$$

The Mie module is capable of computing the spectral single scattering albedo ( $\omega_{LW}$ ), the asymmetry factor ( $g_{LW}$ ) and the normalized extinction coefficient ( $\alpha_{LW}/\alpha_{532}$ ), where  $\alpha_{LW}$  is the spectral extinction coefficient at the LW spectral range, and  $\alpha_{532}$  is the extinction coefficient at 532 nm as provided in López-Cayuela et al. (2023), for each atmospheric layer. Moreover,  $\alpha_{LW}/\alpha_{532}$  also distinguishes between Df and Dc modes. Table 1 shows the input parameters used in the LW spectral range module as well as the data source.

#### 2.2 Dust radiative effect and heating rate estimation

The dust-induced DRE, simulated either at the BOA or the TOA, is defined as in López-Cayuela et al. (2025) (see there Eq. 1). In particular, the atmospheric DRE ( $DRE^{ATM}$ ) is computed as the difference between the DRE at TOA ( $DRE^{TOA}$ ) and that at BOA ( $DRE^{BOA}$ ), that is,  $DRE^{ATM} = DRE^{TOA} - DRE^{BOA}$ . In general, all those quantities are denoted as  $DRE_j^i$ , where i stands for TOA and BOA, and j is the spectral band where DRE is calculated, i.e. j = LW, and NET (SW+LW). All SW magnitudes were previously obtained in López-Cayuela et al. (2025). Both hourly  $DRE_j^i$  for SW and LW were computed for solar zenith angles (SZA) < 90°, since GAME calculates those fluxes only during daytime.

As in López-Cayuela et al. (2025), both the hourly- and daily-averaged  $DRE_{LW}^i$  is calculated. In the SW range, the daily averages were computed as the mean (over 24 hours) of the number of daytime hourly values, as SW fluxes during night-time are zero, unlike those in the LW range. Therefore, night-time hourly LW fluxes were assumed to be equal to the mean value of the daytime LW ones, and hence the daily  $DRE_{LW}$  was obtained from averaging those day-time and night-time-derived hourly (over 24 hours)  $DRE_{LW}$  values. Similar procedure has been applied by other authors (di Sarra et al., 2011; Meloni et al., 2015; Sicard et al., 2022).

Moreover, the fine-to-total (Df/DD) ratio (ftr) of the hourly-averaged  $DRE_j$  ( $ftr\_DRE_j$ , being j = LW, NET) is computed. In addition, a linear fitting analysis of this variable is performed over time, thus obtaining the slope of this linear fitting ( $\delta DRE_j$ ), which serves as an indicator of the temporal rate of the relative contribution of Df particles to the  $DRE_j$ . The dust radiative efficiency ( $DREff_j$ ) is also obtained from the slope of the linear fitting (forced to zero) of DRE values as a function of the dust optical depth at 532 nm (DOD<sup>532</sup>) along the event.

Following the same methodology as in López-Cayuela et al. (2025), differences in the dust-induced DRE (ΔDRE) as obtained from the two approaches are computed as follows:

$$\Delta DRE_{i} = DRE_{i}^{(I)} - DRE_{i}^{(II)},\tag{3}$$

where  $DRE_i^{(I)}$  is the contribution to DRE of Df and Dc particles in each spectral range (i.e., j = LW, NET), that is,

$$DRE_i^{(I)} = DRE_i^{DD} = DRE_i^{Df} + DRE_i^{Dc}, \tag{4}$$

and  $DRE_i^{(II)}$  is the contribution of the total dust as a whole, that is,

$$DRE_{i}^{(II)} = DRE_{i}^{total}. (5)$$

Moreover, the relative differences ( $\Delta^{rel}DRE$ ) between the two approaches were calculated as:

$$\Delta^{rel} DRE_j(\%) = 100 \frac{(DRE_j^{(I)} - DRE_j^{(II)})}{DRE_j^{(II)}}$$
(6)

Furthermore, a statistical analysis based on the relevant percentiles (P), e.g. P(25), P(50) (i.e. median), and P(75), of both  $\Delta DRE$  and  $\Delta^{rel}DRE$  datasets was conducted to evaluate the significance of the discrepancies between the two methodologies.

Finally, it should be noted that aerosols predominantly exhibit a net cooling effect resulting from negative radiative forcing estimates, due to their inherent capacity to scatter solar radiation. However, certain aerosol types such as mineral dust are also

able to absorb radiation to a greater or lesser degree, even likely leading to an opposite effect. Consequently, dust can induce heating in the specific atmospheric layers, despite the potential net cooling effect observed for the overall atmospheric column (Pilewskie, 2007). The aerosol heating rate (AHR, K day<sup>-1</sup>) is defined as the radiatively aerosol-induced rate of the temperature change in time  $\left(\frac{\Delta T}{\Lambda T}\right)$  within a layer of the atmosphere. For a plane-parallel geometry, it can be expressed as follows:

$$AHR(z) = \frac{\Delta T(z)}{\Delta t} = -\frac{g}{c_{pd}} \frac{\Delta F(z)}{\Delta p(z)},\tag{7}$$

where g is the gravity acceleration (9.81 m s<sup>-1</sup>),  $c_{pd}$  is the specific heat of dry air at constant pressure (p) (1005 kJ kg<sup>-1</sup> K<sup>-1</sup>),  $\Delta p$  is the difference of the atmospheric pressure between two layers ( $\Delta p > 0$ ), and  $\Delta F(z)$  represents the corresponding vertical difference in the flux (F(z)) (being  $\Delta z < 0$ ), which is defined as

$$F(z) = \left(F_d^{\downarrow}(z) - F_d^{\uparrow}(z)\right) - \left(F_0^{\downarrow}(z) - F_0^{\uparrow}(z)\right),\tag{8}$$

where  $F_d$  and  $F_0$  denote the solar radiative flux (W m<sup>-2</sup>) as computed by GAME, with and without dust presence, respectively. The arrows indicate whether the fluxes are downward ( $\downarrow$ ) or upward ( $\uparrow$ ).

#### 175 3. Results and discussion

Comprehensive details describing the dust outbreak in overall, and regarding the methodology applied to derive dust optical and microphysical properties from polarized lidar measurements, are reported in López-Cayuela et al. (2023). The DRE analysis in the SW range can be found in López-Cayuela et al. (2025).

#### 3.1 Dust radiative and microphysical properties in the LW range

Figure 1a shows the hourly LST at ARN station, for instance, during the dust outbreak period, where red dots represent the coincident values with lidar measurements when the DRE<sub>LW</sub> can be calculated. Data from all the five Iberian lidar stations can be found in the SM (Figure S1). Therefore, as compared to the SW study (López-Cayuela et al., 2025), from 18% to 45% less DRE<sub>LW</sub> data were available to be analysed. The lidar stations mostly affected by this lack of LST data were ARN and EVO. Regarding the LST results, except for several days with data unavailability, the diurnal LST cycle is nicely visible at the stations. The maximum values ranged from approximately 28 °C (ARN, EVO, BCN) to 32 °C (GRA and TRJ) without significant changes over time (less than 0.02 °C). The maximum night/day difference ranged from approximately 18 °C (BCN) to 30 °C (TRJ).

In Figure 1b,  $r_g$  and  $\sigma_g$  are represented as a function of time, and split into the fine and coarse modes, during the period for ARN station (Figure S2 in the SM shows the same for the rest of the Iberian lidar stations). Those values are obtained from the AERONET  $r_v$  and  $\sigma_v$  (see Eq. 1). A linear fitting of those values over time is also performed. The episode-averaged value and the slope of the linear fitting ( $\gamma$ ) are shown in Table 2. For the fine mode, the mean  $r_g$  ( $\sigma_g$ ) value over the dust episode ranged from 0.076 to 0.093  $\mu$ m (from 0.613 to 0.624  $\mu$ m) at the southern stations (ARN, GRA and EVO). For TRJ and BCN, those values were lower, ranging from 0.059 to 0.067  $\mu$ m (0.552 to 0.575  $\mu$ m). It means that the fine particles were, on average, 10-30% smaller at TRJ and BCN than at the southern stations. Regarding the  $\gamma$  values found, they were positive for GRA, TRJ and BCN, and negative for ARN and EVO. However, that increase/decrease on time was no significant, as it was less than 1%  $\mu$ m day-1 at every station. Therefore, by examining each station individually, the size of the fine particles did not vary considerably throughout the episode. These results are consistent with those obtained by Sicard et al. (2022) for BCN, showing a value of 0.7%  $\mu$ m day-1 during a summer Saharan dust outbreak in 2019.

For the coarse mode, the lowest (highest) mean  $r_g$  value over the dust episode was found at ARN (TRJ), showing a value of 0.471 (0.878). Regarding the  $\sigma_g$ , the lowest (highest) value was found at the same stations, being 0.585  $\mu$ m (0.653  $\mu$ m). For the rest of the stations,  $r_g$  ( $\sigma_g$ ) ranged from 0.529 to 0.584  $\mu$ m (from 0.592 to 0.642  $\mu$ m). Indeed, a value of 6 was found for the episode-averaged coarse-to-fine  $r_g$  ratio at the southern stations, being higher than 10 at TRJ and BCN. Similarly to the fine mode, those values are no significant (lower than 2%  $\mu$ m day<sup>-1</sup>) except for BCN, reaching almost 7%  $\mu$ m day<sup>-1</sup>. This

increase in the geometric radius of the coarse particles at BCN has been observed previously where the coarse dust  $r_g$  increased during a summer Saharan dust outbreak in 2019 at a rate of +9%  $\mu$ m day<sup>-1</sup> (Sicard et al., 2022). A possible explanation for this phenomenon was provided in the aforementioned study. Briefly explained, when the transport of mineral dust occurs over polluted regions with high humidity conditions, not only anthropogenic inorganic acids can be adsorbed onto the dust surface, forming hygroscopic salt compounds that coat the dust particles (Abdelkader et al., 2015; Athanasopoulou et al., 2016), but the formation of secondary pollutants is also enhanced (Querol et al., 2019; W. Xu et al., 2020).

Figure 1c shows the episode-averaged optical properties introduced in the GAME Mie module (see Sect 2.1), i.e. α<sub>LW</sub>/α<sub>532</sub>, g<sub>LW</sub> and ω<sub>LW</sub>, for instance, at the ARN station (for the rest of the Iberian lidar stations see Figure S3 in the SM). It should be noted that the LW range stands for 4-50 μm in this work, and those quantities are separated into the fine, coarse and total modes for the five lidar stations. As it can be observed, the most sensitive spectral window for radiative forcing is between 8 and 13 μm. For that reason, the analysis will be performed by averaging the properties in that spectral window, denoted here by δLW. All those quantities were derived for the separated fine and coarse dust contributions, as well as for the overall bimodal distribution (total dust).

Regarding the episode-averaged properties, similar values were found for  $\alpha_{\delta LW}^{fine}/\alpha_{532}$ , with differences lower than 6% in average between the five lidar stations. Episode-averaged  $g_{\delta LW}^{fine}$  and  $\omega_{\delta LW}^{fine}$  at BCN station were from 2 to 10 times greater with respect to those values found in the other stations. However, it should be noted that those values for the fine mode were remarkable lower than those for the coarse mode. Indeed, the episode-averaged fine-to-coarse ratio of  $g_{\delta LW}$  and  $\omega_{\delta LW}$  is approximately 25% and 5%, respectively. In addition,  $\alpha_{\delta LW}^{fine}/\alpha_{532}$  was an order of magnitude lower than for the coarse mode ( $\alpha_{\delta LW}^{coarse}/\alpha_{532}$ ). Therefore, the DRE<sub>LW</sub> should be mostly dominated by the coarse dust mode.

Furthermore, the  $g_{\delta LW}^{coarse}$  and  $\omega_{\delta LW}^{coarse}$  properties were either rather similar or slightly higher than those corresponding to the total dust. That finding agrees with other studies (e.g. Sicard et al., 2014b). The coarse-to-total ratio was 1.0 and 1.2 on average for  $g_{LW}$  and  $\omega_{LW}$ , respectively. However, in some cases,  $\omega_{LW}^{coarse}$  was 1.7-3.0 times higher. The coarse-to-total  $\alpha_{LW}/\alpha_{532}$  ratio was 1.5-2.0 on average reaching up to values of 3.0-7.0 at several times during the dust event. It means that the (separated) coarse dust should produce a larger extinction in the LW range than that for the total dust (considering the bimodal distribution in overall). This hypothesis will be examined in Sect. 4, where the comparison between the two approaches considered in this study (see Sect. 2.2) will be addressed.

# 3.2 Long-wave dust direct radiative effect (DRE<sub>LW</sub>)

## 3.2.1 DRELW at BOA

Figure 2 shows the hourly dust direct radiative effect in the long-wave range (DRE<sub>LW</sub>, W m²) at BOA, TOA and ATM as induced by the Df and Dc particles at the ARN station, as an example. The rest of the stations can be found in the Supplementary Material (SM) (Figures S3-S6) together with the daily DRE<sub>LW</sub> for all the stations. Table 3 shows the episode-averaged DRE<sub>LW</sub> as induced by Dc, Df and DD at the five Iberian lidar stations, and at BOA and TOA. At all stations, DRE<sub>LW</sub> is positive at BOA for both Df and Dc particles, representing a dust-induced warming at BOA as expected. During the most intense part of the episode (from 26 March to 1 April for all the stations, except for BCN, where lasting until 3 April), hourly Dc (Df) DRE<sub>LW</sub> values below +10 (+1) W m² were mostly found. It should be noted that 27 and 31 March stand out at ARN and TRJ, when DOD<sup>532</sup> > 0.80, showing DRE<sub>LW</sub> values that reached hourly values of ~ +20 (~ +2) W m² for Dc (Df) particles. For the other stations, maximum hourly DRE<sub>LW</sub> values are 40-60% lower than those in ARN (see Table 3). During the rest of the dust event, Dc (Df) DRE<sub>LW</sub> was lower than +5 (+1) W m² at all stations. Regarding the daily DRE<sub>LW</sub> (Figures 2c and 2d) for high values of the daily DD DOD<sup>532</sup> (i.e., ≥ 0.50), DRE<sub>LW</sub> <+20 (+2) W m² for Dc (Df) particles were found, when the daily DD DOD<sup>532</sup> < 0.50 DRE<sub>LW</sub> showed values < +10 (+0.8) W m² for Dc (Df) particles.

Looking at the entire episode, the DRE<sub>LW</sub> averaged values (\overline{DRE\_{LW}}\) ranged from +2.6 to +6.5 (from +0.2 to +0.4) W m<sup>-2</sup> for Dc (Df) particles, as shown in Table 3. It is worth noting that, as observed in L\u00f3pez-Cayuela et al. (2023), aerosol load exhibited very high variability across all stations throughout the study period. Consequently, the standard deviation (std) is considerably high (see Table 3). As it can be observed, the contribution of fine particles to DD DRE<sub>LW</sub> is one order of magnitude lower than

- that of the Dc particles, showing δDRE<sub>LW</sub> values ranging from -2.57 (BCN) to +0.65 (EVO) % day<sup>-1</sup> (Table 3). The mean ftr\_DRE<sub>LW</sub> values for the whole episode were from 6 to 13%, but it should be noted that the maximum hourly ftr\_DRE<sub>LW</sub> ranged from around 10% (GRA) to 41% (BCN). Indeed, between 10% and 15% of the hourly ftr\_DRE<sub>LW</sub> values exceed the mean (± std) of those values for the whole dust episode. Therefore, the Dc DRE<sub>LW</sub> is the one mostly contributing to the DD DRE<sub>LW</sub>. As expected, Df DREff<sub>LW</sub> is also much smaller than Dc DREff<sub>LW</sub> at all the stations (Table 3). Particularly, Df DREff<sub>LW</sub> shows values near to +5 W m<sup>-2</sup> τ<sup>-1</sup>, whereas Dc DREff<sub>LW</sub> ranges from +24 to +37 W m<sup>-2</sup> τ<sup>-1</sup>.
- Performing a comparison analysis of the DRE<sub>LW</sub> obtained in this work with other studies can be challenging as DRE<sub>LW</sub> depends on multiple factors (e.g. fine-to-coarse r<sub>g</sub> ratio, DOD, ftr\_DOD, LST), being able to vary from one dust event to another. Moreover, the DRE<sub>LW</sub> is also highly dependent on the dust layer height (Dufresne et al., 2002; Sicard et al., 2022). Thus, although the results of the present study may agree with several previous works performed for mineral dust in the infrared range over IP stations (e.g., Sicard et al., 2014b, 2022; Granados-Muñoz et al., 2019; Bazo et al., 2023), differences with other studies can be expected. In particular, Sicard et al. (2022) studied the LW direct radiative effect as induced by a summer dust outbreak in 2019 over BCN, considering the same approach as in this work (contribution of the Df and Dc modes separately). That work showed DREff<sub>LW</sub> values of +44.3 (+5.3) W m<sup>2</sup> τ<sup>-1</sup> for Dc (Df) particles. In the present study, similar Df DREff<sub>LW</sub> values were obtained, unlike the Dc DREff<sub>LW</sub> that were almost 50% lower. Variations in this outcome likely arise from the strong sensitivity of LW radiative forcing simulations to aerosol load, coarse-mode particle radius, refractive index, vertical distribution, LST, and surface albedo (Sicard et al., 2014a), thereby accounting for discrepancies with previous studies. For instance, considerable higher LST values were found in Sicard et al. (2022), largely attributed to a concurrent heatwave during the Saharan dust outbreak, with night-time LST values greater than 15°C and maximum daytime LST of 45°C.

#### 3.2.2 DRELW at TOA and in ATM

- Similar to the BOA analysis, DRE<sub>LW</sub> at TOA is positive, representing also a dust-induced heating (Figs. 2a and 2b). However, the magnitude is much lower (vs. DRE<sub>LW</sub> at BOA). In particular, the maximum hourly DRE<sub>LW</sub> values were 2-3 times lower at TOA than at BOA, and they were found at TRJ and BCN, where the dust plume reached higher altitudes (> 6 km) (López-Cayuela et al., 2023). Those specific maximum DRE<sub>LW</sub> values were +10.0 and +6.4 (+0.5 and +0.3) W m², respectively, at TRJ and BCN for Dc (Df) particles, that is, between 2-5 times greater than for the rest of the stations (Table 3). The daily DRE<sub>LW</sub> at TOA (Figure 2c) shows values < +6.0 (+0.3) W m² for Dc (Df) particles for high daily DD DOD<sup>532</sup> (≥ 0.50). For low and moderate daily DD DOD<sup>532</sup> (< 0.50), DRE<sub>LW</sub> decreased to values lower than +2.0 (+0.1) W m² for Dc (Df) particles. Regarding the mean DRE<sub>LW</sub> values as averaged over the whole event, the Dc (Df) DRE<sub>LW</sub> values ranged from +0.9 to +2.3 (from +0.04 to +0.10) W m² for all the stations (Table 3). As shown in Sect. 3.2.1, aerosol load exhibited a very high variability across all stations throughout the study period. Consequently, the standard deviation (std) is considerably high (Table 3).
- The mean ftr\_DRE<sub>LW</sub> for the entire period varied approximately from +4 to +8% at all the stations except BCN, and slightly increased or decreased over time, depending of those stations (Table 3), but no significant impact was observed (δDRE<sub>LW</sub> varied from around -0.5 to +0.4 % day¹). However, at the BCN station, ftr\_DRE<sub>LW</sub> values of around +12% with δDRE<sub>LW</sub> ~ -2.8 % day¹ were found. In addition, note that ftr\_DRE<sub>LW</sub> reached maximum values up to 41%, being the Df contribution rather relevant. Indeed, between 8% and 15% of the hourly ftr\_DRE<sub>LW</sub> exceed the corresponding episode-averaged values (± standard deviation). Since similar results were found at BOA, this agrees with the findings reported by other authors supporting that DRE<sub>LW</sub> is primarily dominated by the contribution of Dc particles (e.g., Sicard et al., 2022).
  - As expected, DREff<sub>LW</sub> is much smaller at TOA than at BOA (Table 3). In particular, DREff<sub>LW</sub> for Dc (Df) particles is overall nearly 4 (7) times lower at TOA (vs. at BOA) over the southern stations (ARN, GRA and EVO), and 2 (3) times lower at the rest. Additionally, the DREff<sub>LW</sub> at TOA for the fine dust component is much smaller than for the coarse dust (Table 3). For all the stations, Df DREff<sub>LW</sub> is lower than +2 W m<sup>-2</sup>  $\tau$ <sup>-1</sup>, and Dc DREff<sub>LW</sub> ranges from +5-7 (at the southern stations) to +12-14 W m<sup>-2</sup>  $\tau$ <sup>-1</sup> (on TRJ and BCN). Thus, Dc DREff<sub>LW</sub> is around 3-10 times greater than Df DREff<sub>LW</sub>.
  - By comparing with other studies (e.g., Granados-Muñoz et al., 2019; Sicard et al., 2022), and regarding the southern stations, similar results for  $DRE_{LW}$  and  $DREff_{LW}$  are found. However, those parameters are 2-4 times greater at TRJ and BCN with respect to those previous studies. The difference in Dc  $DRE_{LW}$  may be explained by the finding of Sicard et al. (2014b), who

- reported that Dc DRE<sub>LW</sub> exhibit little variations when the aerosol optical depth is kept constant. Indeed, Sicard et al. (2014b) and Dufresne et al. (2002) demonstrated that DRE<sub>LW</sub> is highly dependent on the dust layer heights. In comparison with the heights reached by the dust intrusion as reported by Sicard et al. (2022), the observed differences in DRE<sub>LW</sub> and DREff<sub>LW</sub> could be based on this fact. Indeed, the dust plumes reached higher altitudes, especially at TRJ and BCN, during the dust outbreak examined in this study (> 6 km height; López-Cayuela et al., 2023).
- Finally, results on the dust radiative effect in the atmospheric column are reported in Table 3. For instance, hourly DRE<sub>LW</sub> values in ATM at ARN station are shown in Figures 2a and 2b for illustration. The rest of the stations can be found on the SM (Figs. S3-S6). By examining the DRE<sub>LW</sub> at BOA and TOA, the DRE<sub>LW</sub> at ATM is negative during the entire episode at all stations, as DRE<sub>LW</sub> is lower at TOA than BOA, thus indicating a generalised atmospheric dust-induced cooling. The minimum hourly DRE<sub>LW</sub> at ATM (i.e. the most negative) values are found at ARN and TRJ: -15.8 (-1.6) W m<sup>-2</sup> and -12.7 (-0.9) W m<sup>-2</sup> for Df (Dc) particles, respectively. For the rest of the stations the hourly DRE<sub>LW</sub> at ATM minima (i.e. the most negative values) are 50% lower (vs. ARN and TRJ stations; see Table 3). Regarding the episode-averaged estimates, atmospheric Dc (Df) DRE<sub>LW</sub> ranged from -1.3 to -4.2 (from -0.1 to -0.3) W m<sup>-2</sup>.

#### 3.3 Dust net direct radiative effect (DRENET)

#### 3.1. Relationship between DRE<sub>LW</sub> and DRE<sub>SW</sub>

It is known that the LW range is dominated by the Dc particles, whereas the Df particles induce a more pronounced effect in the SW range. In this work, ftr DRE is less than 12% in the LW range (Sect. 3.2), being 45% in and the SW range (López-Cayuela et al., 2025). This fact is illustrated in Figure 3, where the DRELW with respect to DRESW ratio (DRELW/DRESW, in absolute value) is represented, giving an estimation of the percentage of radiative effect that the LW component represents compared to the SW one. For all the stations, the DRE<sub>LW</sub>/DRE<sub>SW</sub> for Df particles ranges on average 4-8%, at both BOA and ATM. At TOA, the magnitude is lower, showing values of 1-4%. Additionally, the Dc DRE<sub>LW</sub>/DRE<sub>SW</sub> ranges 39-54% at BOA, and 20-50% at both TOA and ATM. Particular mention should be made on the case of TRJ, where DRE<sub>LW</sub>/DRE<sub>SW</sub> for Dc reached values of 76%. According to other studies, similar results are found, being the DRE<sub>LW</sub>/DRE<sub>SW</sub> greater at BOA than at TOA. Particularly for desert dust outbreaks in the Mediterranean basin, daily DRE<sub>LW</sub>/DRE<sub>SW</sub> for total dust of 49-52% and 26-35% were found at BOA and TOA, respectively (di Sarra et al., 2011; Meloni et al., 2015). Sicard et al. (2022), which also performed the study by separating both Df and Dc components, found greater values of DRELW/DRESW for coarse dust (67% at BOA, and 60% at TOA). The reason could be attributed to multiple factors thus simulations of LW radiative forcing have demonstrated significant sensitivity to several key parameters, including aerosol load, coarse-mode particle radius, refractive index, vertical aerosol distribution, LST, and surface albedo (Sicard et al., 2014a). Therefore, discrepancies in this variable compared to other studies may be attributed to significant differences in the key parameters described above (see Sect. 3.2.1). Moreover, it is worthy to highlight, as Granados-Muñoz et al. (2019) pointed out, that the results at TOA might not be directly comparable to previous studies due to discrepancies in vertical resolutions within the GAME model for the SW and LW ranges above 4 km, potentially resulting in numerical artefacts in the derived outcomes.

#### 330 3.3.2. DRE<sub>NET</sub> at BOA

By looking at the results, overall,  $DRE_{NET}$  is negative at BOA for all the stations, indicating a dust-induced net cooling effect. Figures 4a and 4b shows the hourly Df and Dc  $DRE_{NET}$ , respectively, at TOA, BOA and ATM in ARN, as an example. Results for the rest of stations are shown in the SM (Figs. S7-S10). Moreover, the daily Df and Dc  $DRE_{NET}$  at BOA for all the stations considered in this study is shown in Figure 4c. Table 4 shows the episode-averaged dust radiative effect in the net range ( $\overline{DRE_{NET}}$ , in W m<sup>-2</sup>) at BOA, TOA and ATM as induced by Df, Dc and DD at the five Iberian lidar stations.

Dc (Df)  $\overline{DRE_{NET}}$  ranges from -2.0 to -6.0 W m<sup>-2</sup> (from -2.9 to -5.7 W m<sup>-2</sup>). Note that those values are rather similar for Dc and Df particles, i.e. both dust components produce on average a similar net cooling at BOA. The daily DRE<sub>NET</sub> (Figure 4c) showed values from -13.1 to -15.5 W m<sup>-2</sup> (from -14.2 to -20.6 W m<sup>-2</sup>) for Dc (Df) particles during days with high daily DD DOD<sup>532</sup> (< 0.50). During days with moderate and low daily DD DOD<sup>532</sup> (< 0.50), DRE<sub>NET</sub> is always lower than -4.8 W m<sup>-2</sup> (-10.8 W

- m<sup>-2</sup>) for Dc (Df). These results slightly agree with a few findings in the literature, reporting daily DD DRE<sub>NET</sub> ranged from -14.6 to -64.0 W m<sup>-2</sup> (Di Sarra et al., 2011; Meloni et al., 2015; Valenzuela et al., 2017). By definition, the DRE<sub>NET</sub> is the sum of their SW and LW components. Therefore, those observed differences might be related to the varying balance between the DRE<sub>SW</sub> (negative) and DRE<sub>LW</sub> (positive). The maximum hourly DRE<sub>NET</sub> was found at ARN station, showing values of -50.4 (Dc) and -43.0 (Df) W m<sup>-2</sup>. For the rest of the stations, the maximum DRE<sub>NET</sub> are lower (in absolute value) than those at ARN (20-60% and 30-65% for Dc and Df, respectively).
  - The impact of fine particles to DD DRE<sub>NET</sub> is mainly due to their dominating contribution in the SW (vs. LW) range, as ftr DREsw was estimated to be around 40% for all the stations (López-Cayuela et al., 2025), meanwhile their LW contribution is between 6% (TRJ) and 13% (BCN) only (ftr DRE<sub>LW</sub>, see Table 3). Indeed, the ftr DRE<sub>NET</sub> values are 45-50%, close to those obtained in the SW for all the stations (see Table 4 in López-Cayuela et al., 2025).
- Moreover, DREff<sub>NET</sub> values at BOA for the Df particles ranged from -128 to -175 W m<sup>-2</sup>  $\tau^{-1}$ , being approximately twice the Dc DREff<sub>NET</sub> (see Table 4). In addition, the DD DREff<sub>NET</sub> showed values from -78 to -114 W m<sup>2</sup>  $\tau^{-1}$  Moreover, Granados-Muñoz et al. (2019) found values of DD DREff<sub>NET</sub> approximately 1.5 times lower at GRA station than those reported in this work. Sicard et al. (2022) reported Dc and DD DREff<sub>NET</sub> values approximately 2 and 1.5 times greater, respectively, at BCN station than those found in this work. Differences could be attributed to the radiative balance in DRE between the LW and SW ranges. In this work, the Dc and DD DREff<sub>NET</sub> is reduced by a factor of 1.2 and 1.4, respectively, by counting on the LW contribution, with respect to the Dc and DD DREff<sub>SW</sub> (López-Cayuela et al., 2025). Those reducing factors agree with the findings of Granados-Muñoz et al. (2019) and Sicard et al. (2022), that is, the DD DREff<sub>NET</sub> is 1.1-1.6 times lower (vs. DD DREff<sub>SW</sub>), although the Dc DREff<sub>NET</sub> is reduced by a slightly higher factor of 2.5.

#### 360 3.3.3. DRENET at TOA and ATM

Similarly to the BOA, the DRE<sub>NET</sub> is negative at TOA (Figure 4d), indicating a dust-induced net cooling effect. In addition,  $DRE_{NET}^{TOA}$  values are 20-30% lower (in absolute units), overall, representing a less pronounced net cooling at TOA with respect to that at BOA. Regarding DRE<sub>NET</sub> on average for the entire episode ( $\overline{DRE_{NET}^{TOA}}$ , Table 4, Dc (Df) DRE<sub>NET</sub> values range from -2.2 to -4.9 W m<sup>-2</sup> (from -2.4 to -4.6 W m<sup>-2</sup>). As it was for  $\overline{DRE_{NET}^{BOA}}$ , note that those values are rather similar for Dc and Df particles, i.e. both dust components produce, on average, a similar net cooling at TOA.

The mean ftr\_DRE<sub>NET</sub> for the entire period was approximately ranging from 49 to 58% between stations (Table 4), with no significant impact observed ( $\delta DRE_{NET} = 0.3 - 0.4 \% \text{ day}^{-1}$ ) at the southern stations, meanwhile slightly higher  $\delta DRE_{NET}$  values of approximately 3-6 % day-1 were found for TRJ and BCN.

The DD DREff<sub>NET</sub> presented nearly 22-34% smaller values at TOA (between -54 and -75 W m<sup>-2</sup>  $\tau$ <sup>-1</sup>) than those at BOA (see Table 5). In particular, DREff<sub>NET</sub> for Dc particles ranged from -37 to -56 W m<sup>-2</sup>  $\tau^{-1}$ , which are around half of the Df DREff<sub>NET</sub> (i.e., from -91 to -123 W m<sup>-2</sup>  $\tau^{-1}$ ). It is important to note (as highlighted in Sect. 3.3.2) that the results obtained at TOA may not be directly comparable with those of previous studies due to differences in the vertical resolution of the GAME model in the SW and LW spectral ranges above 4 km, which could introduce numerical artefacts in results (Granados-Muñoz et al., 2019).

Finally, a dust-induced atmospheric net warming effect can be derived as DRE<sub>NET</sub> is positive at ATM. Those results on the dust direct radiative effect in the atmospheric column are reported in Table 4. For illustration, the hourly DRE<sub>NET</sub> values at ATM at ARN station are shown for Df and Dc particles, respectively, in Figures 4a and 4b (results for the rest of stations are shown in Figures S7-S10 of the SM). The maximum hourly DRE<sub>NET</sub> values at ATM are found at ARN and GRA, showing values of +5.8 (+1.3) W m<sup>-2</sup> and +3.5 (+5.1) W m<sup>-2</sup> for Df (Dc) particles, respectively. For the rest of the stations the hourly DRE<sub>NET</sub> maxima at ATM are lower than +2.6 and +1.5 for Df and Dc particles, respectively (Table 4).

# 3.4. Differences in DRE<sub>LW</sub> and DRE<sub>NET</sub> as estimated using different approaches

Following the approach applied by López-Cayuela et al. (2025) for the SW range, the differences in DRELW and DRENET at all the stations were examined as obtained from the two approaches described in Sect. 2.2.

Relative differences in the LW range (Δ<sup>rel</sup>DRE<sub>Lw</sub>; see Eq. 6) with respect to the classical approach (DRE<sub>Lw</sub><sup>(II)</sup>, see Eq. 5) are presented in Figure 5 as a function of SZA (highlighting the DD DOD<sup>532</sup> dependence). The entire dataset was considered, covering the period from 25 March to 7 April 2021 at all five Iberian lidar stations. No clear correlation was found between Δ<sup>rel</sup>DRE<sub>Lw</sub> and both SZA and DD DOD<sup>532</sup>. At BOA (TOA), mean Δ<sup>rel</sup>DRE<sub>Lw</sub> values of approximately +8.5% (+6.5%) were obtained, although a relatively large standard deviation was observed (~25-27%, see Table 5). Indeed, relatively close Δ<sup>rel</sup>DRE<sub>Lw</sub> values are found for SZA < 70° (see Table 5). An analysis of percentiles further revealed consistent patterns at both levels: P(75) around +16%, P(50) in the interval of +0.8 to +1.2%, and P(25) close to -10% independently on SZA (see Table 5, and Figure 5). These results indicate that larger absolute DRE<sub>Lw</sub><sup>(I)</sup> values relative to DRE<sub>Lw</sub><sup>(II)</sup> are predominantly derived when the full dataset is considered. Specifically, 75% of the Δ<sup>rel</sup>DRE<sub>Lw</sub> values are higher than around -10%, with only 25% falling between -10 to +1%. Consequently, estimates in which the contributions of Df and Dc particles are treated separately are found to represent a more pronounced dust-induced warming effect compared to those obtained when total dust is considered as a single category.

Regarding the absolute differences in  $DRE_{LW}$  ( $\Delta DRE_{LW}$ ; see Eq. 3), those computed from the full dataset were found to be approximately 3-4 times larger at BOA than TOA, with mean (std) values of +0.3 (1.3) and +0.1 (0.5) W m<sup>-2</sup>, respectively, which are close to zero. Maximum (minimum)  $\Delta DRE_{LW}$  values of +9.7 (-2.1) W m<sup>-2</sup> and +2.6 (-1.2) W m<sup>-2</sup> were reached at BOA and TOA, respectively.

However, when the dependence of  $\Delta DRE_{LW}$  on  $r_g$  is examined for the fine and coarse dust, a differentiated behaviour can be observed. Figure 6 displays  $\Delta DRE_{LW}$  as a function of  $DRE_{LW}^{(II)}$  at both BOA and TOA, highlighting the dependence on fine  $r_g$ . It was found that, as fine  $r_g$  increases  $\Delta DRE_{LW}$  shifts from negative to positive values; the same behaviour is observed depending on the coarse  $r_g$ . The inflexion point ( $\Delta DRE_{LW} = 0$ ) was estimated at thresholds of approximately 0.1  $\mu$ m for fine  $r_g$  (or 0.7  $\mu$ m for coarse  $r_g$ ; data not shown). As this study focuses on the relevance of fine particles, reference will be made to the threshold related to fine  $r_g$  throughout this section.

For cases with fine  $r_g < 0.1 \, \mu m$  (i.e. for rather small fine dust particles), the use of the dust-mode separation approach resulted in negative  $\Delta DRE_{LW}$  at both BOA and TOA (see Fig. 6). This reveals an underestimation of  $DRE_{LW}$  values for separated dust components, leading to a less pronounced warming effect. Conversely, when fine  $r_g \ge 0.1 \, \mu m$ ,  $\Delta DRE_{LW}$  tended to be positive, resulting in an overestimation of  $DRE_{LW}$  values with respect to the traditional method, and hence in a more pronounced dust-induced warming effect. Those results are aligned with Sicard et al. (2014b), who demonstrated that the radiative forcing produced by aerosols whose size distribution is dominated by the coarse mode is higher than the estimated by the classical approach. In terms of mean values, the most significant differences are found for size distributions with finer particles, since  $\Delta DRE_{LW}$  showed mean (std) values of +3.1 (2.5) W m<sup>-2</sup> and +0.8 (0.8) W m<sup>-2</sup>, at BOA and TOA, respectively. On the other hand, when fine  $r_g \ge 0.1 \, \mu m$ ,  $\Delta DRE_{LW}$  presented mean (std) values of -0.04 (0.58) W m<sup>-2</sup> and -0.03 (0.22) W m<sup>-2</sup> at BOA and TOA, respectively (see Table 5).

By looking at the main percentiles P(75), P(50) and P(25), as computed from the statistical analysis of  $\Delta$ DRE<sub>LW</sub> (see Table 5), the data distribution is nearly symmetrical, with median values closely matching those mean ones for both fine  $r_g$  intervals. This same pattern depending on fine  $r_g$  is observed at both BOA and TOA, though finding lower  $\Delta$ DRE<sub>LW</sub> at TOA. Indeed, P(25) values indicate that 75% of the  $\Delta$ DRE<sub>LW</sub> values are above +2.0 and +0.6 W m<sup>-2</sup> at BOA and TOA, respectively, for cases with fine  $r_g < 0.1 \, \mu m$ , but close to zero for the remaining cases. These discrepancies observed in dependence of the size interval at both BOA and TOA further emphasizes the critical role of particle size in modulating the vertical distribution and net effect of dust radiative forcing.

The same analysis has been performed for the differences in DRE<sub>NET</sub>. Figure 7 shows  $\Delta^{rel}$ DRE<sub>NET</sub> as a function of SZA for all five lidar stations and the whole dataset. In this case, a clear dependence on SZA is observed, originating from the same effect as in the SW range (López-Cayuela et al. 2025). Specifically, larger differences are found for SZA > 70°, although the effect is less pronounced than in the SW range (López-Cayuela et al. 2025), as it is modulated by the contribution of the LW range to the net radiative balance. At TOA and for SZA > 70°,  $\Delta^{rel}$ DRE<sub>NET</sub> values are mostly positive, ranging from approximately - 10% to +65 %, with a mean (std) value of +14.0 (20.0)% (see Figure 7a). For the same SZA range,  $\Delta^{rel}$ DRE<sub>NET</sub> at BOA showed values that ranged from around -35% to +90%, and with a mean (std) value of +12.7 (22.7)% (see Figure 7b). As explained in

López-Cayuela et al. (2025), the significant Δ<sup>rel</sup>DRE<sub>SW</sub> found for SZA > 70° are associated to the intrinsic uncertainty in GAME simulations resulting from the model assumption of a plane-parallel atmosphere, and hence those values should be discarded.

Thus, once disregarding values of  $\Delta^{\text{rel}}DRE_{\text{NET}}$  for SZA > 70°,  $\Delta^{\text{rel}}DRE_{\text{NET}}$  are mostly negative at both BOA and TOA, showing mean (std) values of -4.8 (6.6)% and -8.5 (13.0) %, respectively. This is also corroborated by looking at the percentiles P(25) and P(75), which show values, respectively, of -9.6 and -1.6% at BOA, and -13.4 and -1.2% at TOA (see Table 6). Indeed, those results indicate that 75% of  $\Delta^{\text{rel}}DRE_{\text{NET}}$  values are below around -2 and -1% at BOA and TOA, respectively, with minima of -18.2 and -80%. This represents, as DRE<sub>NET</sub> is negative, a less pronounced net cooling at both BOA and TOA when the Df and Dc contribution is separately (vs. total dust) accounted for and showing larger differences at TOA (vs. BOA).

Finally, Figure 8 shows the differences in DRE<sub>NET</sub> (ΔDRE<sub>NET</sub>) obtained from the two approaches at both BOA and TOA for all five lidar stations. It should be noted that absolute  $\Delta DRE_{NET}$  tend to increase as  $DOD^{532}$  increases. In general,  $\Delta DRE_{NET}$ were mostly close to zero at lower DOD (< 0.2), and increased somewhat at moderate/high-dust-load conditions (DOD > 0.50). Indeed, those differences in DRE<sub>NET</sub> reached minimum/maximum values of -6.4/+6.4 and -10.4/+2.3 W m<sup>-2</sup>, respectively, at both BOA and TOA for SZA < 70°, showing positive (and close to zero) mean (std) values of +0.5 (1.5) and +0.2 (1.4) W m <sup>2</sup>. This can be corroborated by looking at the percentiles: 75% of  $\Delta DRE_{NET}$  values are mostly positive (i.e., P(25) = +0.2 and +0.1 W m<sup>-2</sup> at BOA and TOA, respectively). Table 6 shows all those values. Moreover, it should be noted that a differentiated behaviour is observed around a DRE<sub>NET</sub> threshold of -20 W m<sup>-2</sup>. In particular, when DRE<sub>NET</sub>(III) > -20 W m<sup>-2</sup>, similar low mean  $\Delta DRE_{NET}$  values are obtained at BOA (+0.5 ± 2.5 W m<sup>-2</sup>) and TOA (+0.5 ± 0.5 W m<sup>-2</sup>). However, for  $DRE_{NET}^{(II)} \le -20$  W m<sup>-</sup>  $^{2}$ , corresponding to higher dust load conditions,  $\Delta DRE_{NET}$  show positive mean values at BOA (+0.6 ± 0.8 W m<sup>-2</sup>) and negative mean values at TOA ( $-2.0 \pm 2.6 \text{ W m}^2$ ). Moreover, as shown in Table 6, 75% of  $\Delta DRE_{NET}$  are above  $\pm 0.3 \text{ W m}^2$  at BOA, and below -0.3 W m<sup>-2</sup> at TOA. Overall, these results would indicate a less pronounced net cooling at BOA in contrast of a more pronounced net cooling at TOA when the separation contribution of the Df and Dc particles (vs. total dust) is regarded under high dusty conditions. This highlights a potential modulation of the dust impact in the atmosphere, being potentially able to produce an atmospheric net cooling (contrary to what stated in Sect. 3.3). However, those final remarks should be carefully regarded as only 8% of those examined DRE<sub>NET</sub> profiles correspond to DOD values greater than 0.5.

## 3.5 Aerosol heating rate

The vertical aerosol heating rate (AHR) has been computed in the SW and LW range (see Sect. 2.2, Eq. 7) for all the dust components (DD, Df, Dc). Maxima of the hourly AHR ( $AHR^{max}$ , K day<sup>-1</sup>) for the entire dust episode at each lidar station together with the episode-averaged of those values and their corresponding heights are shown in Table 7. Regarding the SW range, the AHR<sub>SW</sub> is predominantly positive, with maximum values within the dust layer, indicating a warming effect in the atmosphere. On the contrary, near the surface, AHR<sub>SW</sub> are mostly negative (cooling effect). As the fine-to-total AHR ratio in the SW ( $ftr\_AHR_{SW}$ ) is nearly constant for all stations (around 30% in the dust layer; see Table 7), the discussion is focused on DD AHR<sub>SW</sub> (for the sake of reading, Figure S11 of the SM shows the AHR<sub>SW</sub> at the five Iberian lidar stations). As stated in several works, the AHR<sub>SW</sub> is linked to the vertical distribution of the dust extinction, and its magnitude increases with the DOD (Perrone et al., 2012; Meloni et al., 2015; Peris-Ferrús et al., 2017). An extensive study of the vertical dust extinction distribution can be found in López-Cayuela et al. (2023, 2025).

To summarize, the dust plume appeared below 3 km at the beginning of the dust event (25–26 March) over the southern stations (ARN, GRA, EVO). On 27 March, enhanced instability lifted the plume up to 6 km. The strongest intrusion occurred on 29–31 March, with dust extending from the surface to ~7 km height. From 1 April, the plume weakened and descended to ~3 km (see Figure 1 in López-Cayuela et al., 2025). Regarding the AHR<sub>SW</sub>, the maxima varied from ~ 0.3-1.0 K day<sup>-1</sup> at the beginning of the episode, reached ~ 3 K day<sup>-1</sup> on 31 March, and showed values of ~0.1-0.3 K day<sup>-1</sup> at the end of the episode, at altitudes of 3-5 km. At the central station (TRJ), the dust plume was first detected below 4 km on 26 March, ascending to 10 km later that day. In the following days, plume top heights fluctuated between 6–8 km height, occasionally reaching 10 km. The strongest intrusion occurred also on 29–31 March. From 1 April, the plume subsided from 8 to 4 km with reduced intensity (see Figure 1 in López-Cayuela et al., 2025). The maxima AHR<sub>SW</sub> varied at altitudes of 4-6 km from ~ 0.4-0.7 K day<sup>-1</sup> at the beginning of the dust outbreak, peaking ~ 2 K day<sup>-1</sup> on 31 March, to ~ 0.1 K day<sup>-1</sup> at the end of the episode. At BCN station,

the dust plume was persistently stratified and less intense than at the other lidar sites, though plume tops reached 10 km height on several occasions. On 28 March, dust was confined to 2–3 km. On 29 March, the structure became more complex, with two distinct layers (at 2–3 and 9–10 km) in the morning and three layers (at 1–2, 4–7, 8–10 km) later in the day. A similar stratification persisted in the following days, with maximum incidence on 1 April. Afterwards, the dust intrusion gradually weakened until 6 April (see Figure 1 in López-Cayuela et al., 2025). The maxima AHR<sub>SW</sub> varied at altitudes of 2-6 km, from ~ 0.1-0.3 K day¹ until 1 April, when peaked to ~ 0.7 K day¹, to finally decreased to ~0.1 K day¹ at the end of the episode. Finally, by averaging over the whole dusty period the maximum hourly DD AHR<sub>SW</sub>, those values peaked at corresponding mean altitudes between 2.0 (BCN) and 5.3 (GRA) km height, and ranged from +0.20 to +0.50 K day¹ (see Table 7).

Regarding the LW range, AHR<sub>LW</sub> is mainly negative. As the fine mode contribution to the DD AHR<sub>LW</sub> is low (*ftr\_AHR<sub>LW</sub>* <16%; see Table 7), the discussion will focus on DD AHR<sub>LW</sub> (the vertical distribution of DD AHR<sub>LW</sub> can be found in Figure S12 of the SM). The absorption of SW radiation by the dust layer led to the emission of LW radiation in all directions, resulting in a negative AHR<sub>LW</sub> (cooling effect). Indeed, by looking at the vertical AHR structure, the AHR<sub>LW</sub> profiling usually peaks at lower altitudes than those for AHR<sub>SW</sub>, being maxima below the dust layer. This behaviour was also found in other studies (e.g., Sicard et al., 2014a). As stated in Meloni et al. (2015), when the dust intrusion is structured in several layers, the AHR<sub>LW</sub> sign changes from negative to positive below the most dust-loaded layer, depending on the absorption of the lowermost layer. Generally, when the extinction coefficient indicates a significant aerosol load in the lowermost atmosphere, AHR<sub>LW</sub> predominantly remains negative, revealing a phenomenon linked to thermal emissions within the layer itself. Conversely, AHR<sub>LW</sub> tends to be positive when the aerosol extinction in the lowermost atmosphere is negligible. In such instances, the primary dust layer induces a LW heating effect beneath it, attributable to the absorption of local thermal radiation emitted by the dust layer (Meloni et al., 2015). The maximum (negative) values of the hourly DD AHR<sub>LW</sub> were found at the days of the maximum incidence at altitudes below the dust layer (López-Cayuela et al., 2023), ranging from -0.12 (EVO) to -0.85 (TRJ) K day<sup>-1</sup>. The period-averaged values of the maximum (negative) hourly DD AHR<sub>LW</sub> for each station and spectral range are shown in Table 7, ranging from -0.04 K day<sup>-1</sup> at 1.7 km (BCN) to -0.12 K day<sup>-1</sup> at 1.6 km (TRJ).

500 It should be noted that for both SW and LW ranges results on AHR agree with those found in previous studies for mineral dust. Indeed, they reported hourly AHR<sub>SW</sub> and AHR<sub>LW</sub> values ranging, respectively, from +0.30 to +3.80 K day<sup>-1</sup>, and from -0.30 to -0.70 K day<sup>-1</sup> (Sicard et al., 2014a; Meloni et al., 2015; Peris-Ferrús et al., 2017; Valenzuela et al., 2017; Bazo et al., 2023).

Figure 9 shows the vertical AHR<sub>NET</sub> (K day<sup>-1</sup>) for DD particles at the five Iberian lidar stations along the dust event, together with an example of hourly AHR<sub>SW</sub> and AHR<sub>LW</sub> profiling at each site. As stated for the AHR in the SW range, similarly ftr\_AHR<sub>NET</sub> is also nearly constant for all stations (nearly 30% within the dust layer; see Table 7). AHR<sub>NET</sub> profiles show negative values (net cooling) in the lowermost atmosphere during the dusty period, as both LW and SW contributions are negative, being consistent with the negative DRE<sub>NET</sub> at BOA (Sect. 3.3.1). In contrast, positive AHR<sub>NET</sub> (net warming) dominates within the dust layer, where the (positive) AHR<sub>SW</sub> prevails. Figure S13 of the SM shows the vertical distribution of the LW-to-net AHR ratio (AHR<sub>LW</sub>/AHR<sub>NET</sub>; %). It can be seen that the LW contribution to AHR<sub>NET</sub> is generally < 10% inside the dust layer, but nearly all of it occurs below, and to a lesser extent above, the layer (see Fig. S13 in the SM). Overall, AHR<sub>NET</sub> is positive in the most dust-loaded layer and negative below and above it, typically between 2–4 km height and occasionally above 8 km, being consistent with López-Cayuela et al. (2023). Although AHR<sub>SW</sub> dominates, AHR<sub>LW</sub> remains relevant as it modulates the strength of the net effect. As expected, the maxima values are found on day of maxima DOD, within the dust layer, and reaching values from +1.83 K day<sup>-1</sup> (ARN) to +0.89 K day<sup>-1</sup> (BCN). The dust period-averaged DD AHR<sub>NET</sub> showed values ranging from +0.22 K day<sup>-1</sup> around 5 km (BCN) to +0.44 K day<sup>-1</sup> around 3 km (TRJ) (see Table 7).

#### 4. Summary and conclusions

This work is complementary to the research conducted by López-Cayuela et al. (2023, 2025), with the aim of introducing thus the closure study about the vertical radiative impact of an intense and long-lasting Saharan dust outbreak over the Iberian Peninsula in springtime 2021. In this work, the temporal variation of the DRE in the LW range and net DRE was estimated, separating the Df and Dc contributions. For that purpose, lidar observations in five Iberian stations: El Arenosillo/Huelva

(ARN), Granada (GRA), Torrejón/Madrid (TRJ), and Barcelona (BCN) in Spain, and Évora (EVO) in Portugal, were used. The key findings are summarised below.

The availability of DRE<sub>LW</sub> computations was reduced by 18–45% compared to the DRE<sub>SW</sub> reported in the accompanying article due mainly to limited LST measurements, particularly at ARN and EVO. Despite some data gaps, the diurnal LST cycle was clearly observed, with maximum values between 28 °C and 32 °C and minor temporal variability (< 0.02 °C). The fine geometric median radius r<sub>g</sub> and its standard deviation σ<sub>g</sub> were larger at the southern IP stations (ARN, GRA, EVO) than at TRJ (central IP) and BCN (north-eastern IP), indicating 10–30% smaller fine particles in the latter stations. Temporal trends in r<sub>g</sub> were negligible across stations (< 1% μm day⁻¹), indicating that fine particle size remained stable during the dust outbreak. Although many stations showed stable coarse particle sizes, BCN experienced a significant increase (~ 7% μm day⁻¹), being consistent with previous studies. This may be due to dust interaction with anthropogenic pollutants and humid conditions enhancing particle growth. The coarse mode dominated the LW extinction and scattering processes, as indicated by higher g and ω values, and their coarse-to-total ratios exceeding unity, reaching up to 3-7 for the extinction. The dominant role of coarse dust in the LW spectral range (particularly 8-13 μm) implies that LW radiative forcing estimations should be mainly affected by coarse-mode contributions. These findings align with previous literature and were validated in Section 4.

On the one hand, a dust-induced warming at BOA was consistently observed for both fine and coarse dust particles, with Dc contributing the most. During the highest incidence of the dust episode, maximum hourly DRE<sub>LW</sub> values at BOA reached up to  $+20~W~m^{-2}$  (Dc) and  $+2~W~m^{-2}$  (Df), especially at ARN and TRJ stations. The daily DRE<sub>LW</sub> values were significantly lower when the dust optical depth at 532 nm (DOD<sup>532</sup>) was below 0.50, indicating a direct dependency on dust concentration. Episode-averaged DRE<sub>LW</sub> values ranged from +2.6 to  $+6.5~W~m^{-2}$  for Dc and from +0.2 to  $+0.4~W~m^{-2}$  for Df particles. This shows that Df contributes an order of magnitude less to the LW radiative forcing. The relative contribution of Df ( $fir_{DRE_{LW}}$ ) was 6-13% on average, but it could reach up to 41% during some times, particularly at BCN. Despite these peaks, the Dc component remains the dominant driver of the total dust DRE<sub>LW</sub>. Regarding the radiative efficiency, DREff<sub>LW</sub> values were much higher for Dc (from +24 to  $+37~W~m^{-2}~\tau^{-1}$ ) than Df ( $\sim +5~W~m^{-2}~\tau^{-1}$ ), reinforcing the greater radiative impact of coarse dust

This study also provides a comprehensive assessment of the DRE<sub>NET</sub>, being consistently negative at both BOA and TOA, and hence reflecting a net cooling effect induced by dust. In contrast, a positive DRE<sub>NET</sub> was observed within the atmospheric column, suggesting a dust-driven net warming at this level. Regarding the effect of the Df particles at both BOA and TOA, they contributed a maximum of 12% and 30% to the DRE<sub>NET</sub> in the LW and SW ranges, respectively. The Df DRE<sub>NET</sub> was similar to the Df DRE<sub>SW</sub>, since the Df DRE<sub>LW</sub> is nearly negligible. Indeed, ftr\_DRE<sub>NET</sub> was approximately 45-50% at BOA, and 50-60% at TOA. In addition, the DREff<sub>NET</sub> corresponding to Df particles was almost half of Dc DREff<sub>NET</sub> at both BOA and TOA. Among all stations, ARN exhibited the highest hourly DRE<sub>NET</sub> magnitudes, highlighting the influence of local atmospheric and surface conditions on radiative forcing. The inclusion of the LW component was found to decrease the net radiative efficiency, with reduction factors ranging from 1.2 to 2.5, depending on the dust mode fraction. This underscores the importance of accounting for the SW–LW balance when quantifying the net radiative impact of dust. All these findings contribute to a better understanding of the vertical distribution of dust radiative effects and their implications for regional climate over the IP.

Concerning the vertical distribution of the aerosol heating rates, the peak of AHR<sub>SW</sub> profiling occurred at a higher altitude than that of the AHR<sub>LW</sub> one. Moreover, the AHR<sub>SW</sub> was predominantly positive (warming effect within the dust layer) unlike the AHR<sub>LW</sub>, which was negative (cooling effect). Hence, in the dust layer, AHR<sub>NET</sub> displayed a warming effect. On the contrary, below (until 2-4 km) and above (beyond 8 km) the dust layer, AHR<sub>NET</sub> indicates a cooling effect. Moreover, the ft\_AHR<sub>NET</sub> in the dust layer for all the stations, in average, is nearly 30%, being the contribution of the AHR in the LW range, as maximum, of 10% to AHR<sub>NET</sub>. Despite AHR<sub>SW</sub> dominates leading to a predominant atmospheric warming effect of dust, the relevance of the opposite (cooling) AHR<sub>LW</sub> effect relies on the potential atmospheric modulation by reducing the strength of the warming net impact.

On the other hand, as a novelty in this work, the use of two methodologies to estimate the DRE in each spectral range has been examined in detail. For that purpose, differences in DRE ( $\Delta DRE$ ) by using the combination of the separately computed contribution of the two dust components (Dc, Df) to the DRE (approach 1) against the classical estimation (approach 2,

considering the total dust as a whole) were analysed. Results revealed that the classical approach underestimated the DRE<sub>LW</sub>, with a mean value of nearly +8-9% at both BOA and TOA in terms of the relative differences. Moreover, for cases with rather small fine dust particles, the use of the dust-mode separation approach resulted in negative LW differences at both BOA and TOA, revealing an underestimation of DRE<sub>LW</sub> values by using the separated dust components approach. This leads to a less pronounced warming effect unlike when fine dust radii are larger than 0.1 μm, resulting in a DRE<sub>LW</sub> overestimation with respect to the traditional method, and hence in a more pronounced dust-induced warming effect. These discrepancies observed in dependence of the size interval at both BOA and TOA further emphasizes the critical role of particle size in modulating the vertical distribution of the DRE<sub>LW</sub>, and then affecting the net dust radiative forcing.

Indeed, the dust-induced net effect is strongly affected by the SW range, but modulated by the LW range. DRE<sub>NET</sub> is overestimated, on average, by the classical approach, being the relative differences between approaches of -5% at BOA and -9% at TOA. Moreover, a less pronounced net cooling is obtained at BOA in contrast of a more pronounced net cooling at TOA when the separation contribution of the Df and Dc particles (vs. total dust) is regarded under moderate-high dusty conditions. This highlights a potential modulation of the dust impact in the atmosphere, being potentially able to produce actually an atmospheric net cooling.

The literature consistently indicates that global models underestimate the burden and extent of Dc particles, leading to an underestimation of its LW warming and of its reduced SW cooling effectiveness. By incorporating dust observations, both regional and global direct radiative forcing is altered, improving agreement between simulations and observations. While SW dominates surface cooling, Dc-induced LW contributes to warming at TOA and within the atmospheric column, further conditioning semi-direct cloud effects. Dust size-resolving studies show that computing DRE in bulk overestimates (in absolute terms) SW cooling; when dust components (fine and coarse modes) are separated, Df and Dc contributions partly offset each other, yielding a less negative radiative forcing (less pronounced cooling) at TOA, whereas the net DRE remains strongly negative at BOA. Recent estimates highlight a globally relevant positive LW direct forcing during the industrial era, previously overlooked due to the lack of realistic coarse-mode representation. Therefore, separating fine and coarse dust contributions to DRE is crucial, affecting the TOA radiative forcing sign, BOA attenuation magnitude, atmospheric heating profiles, and model biases linked to underrepresented Dc particles and their LW effects. Thus, the dust radiative impact, and related cloud adjustments, can be wrongly estimated in both sign and magnitude when not considering this dust component separation.

**Data availability.** EARLINET lidar files are available from the EARLINET data portal (https://data.earlinet.org/, last access: 21 December 2021; Pappalardo et al., 2014). The accessibility of these files is limited based on the EARLINET criteria. Part of the data used in this publication were obtained as part of the AERONET network and are publicly available. For additional lidar data or information, please contact the corresponding author.

Author contributions. MÁLC, CCJ, and JLGR conceptualized the study. MÁLC, CCJ, MS, and JLGR were responsible for the formal analysis. MÁLC wrote the original draft of the paper and applied the software. MÁLC, CCJ, MS, and JLGR carried out the investigation. MÁLC, CCJ, MS, VS, MJGM, AC, FTC, JABA, CMP, MJC, ARG, DB, JAG, LAA, and JLGR reviewed and edited the paper. CCJ, MJGM, ARG, and DB were responsible for data curation. CCJ, LAA, AC, and MJC provided the resources. CCJ and JLGR supervised the investigation. All authors have read and agreed upon the published version of the paper.

Competing interest. The contact author has declared that none of the authors has any competing interests.

Acknowledgements. This work was funded by the Agencia Estatal de Investigación (AEI)/Ministerio de Ciencia, Innovación y Universidades (MICIU) and FEDER "Una manera de hacer Europa" (grant PID2023-151666NB-I00, PID2023-149747NB-I00, PID2024-158786NB-C21, PID2024-158786NB-C22, EQC2018-004686-P, and RED2024-153891-E), supported by the University of Granada (the Singular Laboratory programme LS2022-1, and the Scientific Units of Excellence Programme

- grant UCE-PP2017-02), and partially supported by the EU H2020 (ACTRIS GA 871115) and MSCA Staff Exchange (grant 101236396). The PT team is co-funded by national funds through FCT Fundação para a Ciência e Tecnologia, I.P., in the framework of the project UIDB/06107 Centro de Investigação em Ciência e Tecnologia para o Sistema Terra e Energia CREATE. Michaël Sicard acknowledges the support of the European Commission through the REALISTIC project (GA 101086690) and CNES through the projects EECLAT, AOS, and EXTRA-SAT. María-Ángeles López-Cayuela is supported
- by the INTA predoctoral contract programme. María-Ángeles López-Cayuela thanks ATMO-ACCESS for the TNA LIRTASOM ("Lidar data in Radiative Transfer model for dust direct radiative effect estimation and evaluation against solar measurement") project, supported by the European Commission (H2020-INFRAIA-2020-1, grant 101008004). Jesús Abril-Gago thanks the Spanish Ministry of Universities for the grant FPU 21/01436. The BCN team thanks Ellsworth J. Welton for providing the MPL unit at the Barcelona site. Ellsworth J. Welton and Sebastian A. Stewart are warmly acknowledged for
- their continuous help in keeping the MPL systems up to date. The authors gratefully acknowledge the PIs and technical staff of all the lidar and AERONET stations for maintenance support of the instrumentation involved in this work.

**Financial support.** This research has been supported by the Agencia Estatal de Investigación (AEI)/Ministerio de Ciencia, Innovación y Universidades (MICIU) (grant no. PID2023-151666NBI00).

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

Figure 1. a) Hourly land surface temperature (LST, in °C), where the red dots represent the values coincident with lidar measurements; b) AERONET geometric median radius ( $r_g$ , in  $\mu$ m) and standard deviation ( $\sigma_g$ ) for the (Left) fine and (Right) coarse modes, where the dashed lines represent the linear fitting of  $r_g$  over time; and c) Episode-averaged values of the (Left) spectral normalized extinction ( $\alpha_{LW}/\alpha_{532}$ ), (Centre) asymmetry factor ( $g_{LW}$ ), and (Right) single scattering albedo ( $\omega_{LW}$ ), for the fine (blue), coarse (red) and total (yellow) modes. All the plots refer to El Arenosillo/Huelva (ARN) station; for the rest of stations, see the Supplementary Material (SM).

Figure 2. Dust direct radiative effect in the long-wave range (DRE<sub>LW</sub>, W m<sup>-2</sup>) at BOA (purple), TOA (green) and in the atmosphere (ATM, yellow) at the ARN station, for instance, as induced by the a) fine dust (Df), and b) coarse dust (Dc) particles. Daily mean DRE<sub>LW</sub> values at c) BOA and d) TOA for Df (right) and Dc (left) particles at the five lidar stations.

Figure 3. Daily DRE<sub>LW</sub>/DRE<sub>SW</sub> ratio (%; in absolute units) at the five Iberian lidar stations, at BOA (purple), TOA (green), and in 5 ATM (yellow), corresponding to: a) Df particles, and b) Dc particles. The episode-averaged values are also shown on the right of each panel.

Figure 4. The same as Fig. 2, but for the net DRE (DRE $_{NET}$ ).

Figure 5. Relative differences in DRE<sub>LW</sub> (A<sup>rel</sup>DRE<sub>LW</sub>, in %) as obtained from the two approaches (Eq. 6) as a function of SZA at: a) TOA, and b) BOA, for all five lidar stations from 25 March to 7 April 2021. The dependence on DD DOD<sup>532</sup> is shown as a colour-scaled bar at the top.

Figure 6. Differences in DRELW as obtained from the two approaches ( $\Delta DRE_{LW} = DRE_{LW}(^{11}) - DRE_{LW}(^{11})$ , Eq. 3; W m<sup>-2</sup>) as a function of DRELW(^{11}) at: a) TOA, and b) BOA, for all five lidar stations from 25 March to 7 April 2021. The dependence on the fine  $r_g(\mu m)$  is shown as a colour-scaled bar at the top.

Figure 7. The same as Fig. 5, but for DRE<sub>NET</sub> ( $\Delta^{rel}$ DRE<sub>NET</sub>, in %). The vertical dashed black line denotes SZA = 70°.

Figure 8. Differences in DRE<sub>NET</sub> as obtained from the two approaches ( $\Delta$ DRE<sub>NET</sub>=DRE<sub>NET</sub><sup>(I)</sup>-DRE<sub>NET</sub><sup>(II)</sup>; Eq. 3; W m<sup>-2</sup>) at a) TOA, and b) BOA, for all five lidar stations from 25 March to 7 April 2021. The dependence on the DD DOD<sup>532</sup> is shown as a colour-scaled bar at the top. Data for SZA > 70° are highlighted by cross symbols. The vertical dashed line indicates DRE<sub>NET</sub><sup>(II)</sup> = -20 W m<sup>-2</sup>.

Figure 9. Left) Vertical distribution of the net aerosol heating rates (AHR<sub>NET</sub>, K day<sup>-1</sup>) corresponding to dust (DD) particles at the five Iberian lidar stations (from North-East to South-West, by decreasing latitude): a) BCN, b) TRJ, c) EVO, d) GRA and e) ARN stations. (Right) An example of an hourly-averaged AHR profile for the SW (red) and LW (blue) range at each station. These specific profiles are marked on the left panel between white arrows. Solid, dashed and dotted lines refers to DD, Df and Dc AHR<sub>NET</sub>, respectively.

Table 1. Input parameters and radiative properties for the GAME model in the LW spectral range. Note that z denotes the vertical dependence.

|                     | Parameter                                | Database / instrumentation          |  |  |
|---------------------|------------------------------------------|-------------------------------------|--|--|
|                     | SA                                       | 0.017 (Sicard et al., 2014a)        |  |  |
|                     | LST                                      | COPERNICUS                          |  |  |
| Atmosphere and land | Meteorological profiles                  | U.S. std. atmos. + 3h GDAS profiles |  |  |
|                     | Gas concentration profiles               | U.S. std. atmos. + 3h GDAS profiles |  |  |
|                     | Absorption coefficients                  | HITRAN                              |  |  |
|                     | $\alpha_{532}(z)$ (fine, coarse, total)  | Lidar                               |  |  |
|                     | DOD <sup>532</sup> (fine, coarse, total) | Lidar                               |  |  |
| A I .               | g (fine, coarse, total)                  | AERONET                             |  |  |
| Aerosols            | ω                                        | AERONET                             |  |  |
|                     | $r_g$ , $\sigma_g$ (fine, coarse, total) | AERONET                             |  |  |
|                     | Refractive index                         | Krekov (1993)                       |  |  |

Table 2. Episode-averaged median radius  $(r_g, \mu m)$  and standard deviation  $(\sigma_g, \mu m)$  at the five lidar stations: Barcelona (BCN), Torrejón/Madrid (TRJ), Évora (EVO), Granada (GRA) and El Arenosillo/Huelva (ARN) for the fine and coarse modes. The slope of each linear fitting  $(\gamma, \% \mu m \text{ day}^{-1})$  is also shown.

|             |               | ARN    | GRA    | EVO    | TRJ    | BCN    |
|-------------|---------------|--------|--------|--------|--------|--------|
| Fine mode   | $r_g$         | +0.076 | +0.093 | +0.083 | +0.067 | +0.059 |
|             | $\gamma(r_g)$ | -0.5   | 0.6    | -0.4   | +0.4   | +0.8   |
|             | $\sigma_g$    | +0.613 | +0.651 | +0.624 | +0.575 | +0.552 |
| Coarse mode | $r_g$         | +0.471 | +0.584 | +0.529 | +0.878 | +0.578 |
|             | $\gamma(r_g)$ | -0.5   | -2.0   | -0.6   | +2.0   | +6.9   |
|             | $\sigma_g$    | +0.585 | +0.584 | +0.592 | +0.653 | +0.642 |

Table 3. Episode-averaged dust direct radiative effect in the LW range (DRE<sub>LW</sub>, in W m<sup>-2</sup>), and the standard deviation (in brackets), at the BOA and TOA (and ATM) as induced by fine (Dc), coarse (Dc) and total dust (DD) at the five Iberian lidar stations.  $\bar{X}$  indicates the mean value for the whole event (standard deviations are also shown), and  $X^{max}$  refers to the maximal value. The DREff (in W m<sup>-2</sup>  $\tau^{-1}$ ) denotes the DRE efficiency. The ftr\_DRE denotes the hourly Df-to-DD DRE ratio (in %,), showing also the mean, median, minimum (min), and maximum (max) values;  $\delta DRE$  (in % day<sup>-1</sup>) is the slope of the linear fitting analysis of the hourly ftr\_DRE values along time.

|     | LW                 |        | ARN          | GRA          | EVO          | TRJ          | BCN          |
|-----|--------------------|--------|--------------|--------------|--------------|--------------|--------------|
|     |                    | Df     | +0.4 (0.5)   | +0.3 (0.3)   | +0.3 (0.1)   | +0.4 (0.3)   | +0.2 (0.1)   |
|     | $\overline{DRE}$   | Dc     | +4.7 (6.2)   | +3.6 (3.0)   | +3.9 (2.8)   | +6.5 (5.2)   | +2.6 (2.0)   |
|     |                    | DD     | +5.1 (6.7)   | +3.9 (3.2)   | +4.3 (2.9)   | +7.0 (5.5)   | +2.8 (2.1)   |
|     | DRE <sup>max</sup> | Df     | +1.7         | +0.7         | +0.6         | +1.3         | +0.8         |
|     | DKL                | Dc     | +19.8        | +8.1         | +8.1         | +22.7        | +12.3        |
|     |                    | Df     | +5.0 (0.1)   | +4.5 (0.1)   | +5.0 (0.1)   | +5.1 (0.1)   | +4.9 (0.1)   |
| BOA | DREff              | Dc     | +25.6 (0.6)  | +26.6 (1.9)  | +27.3 (1.1)  | +36.8 (0.7)  | +24.3 (1.3)  |
|     |                    | DD     | +19.6 (0.4)  | +20.2 (1.3)  | +20.6 (0.8)  | +27.2 (0.5)  | +19.0 (0.9)  |
|     | $\delta DRE$       |        | +0.41        | -0.10        | +0.65        | -0.05        | -2.57        |
|     |                    | mean   | +10.9 (4.9)  | +6.3 (1.7)   | +9.8 (3.5)   | +5.9 (1.8)   | +13.2 (10.8) |
|     | ftr_DRE            | median | +10.1        | +5.8         | +9.4         | +5.4         | +8.4         |
|     |                    | min    | +4.6         | +3.3         | +3.9         | +3.7         | +2.5         |
|     |                    | max    | +31.9        | +9.6         | +17.5        | +12.9        | +40.7        |
|     |                    | Df     | -0.3 (0.4)   | -0.3 (0.2)   | -0.3 (0.1)   | -0.3 (0.2)   | -0.1 (0.1)   |
|     | DRE                | Dc     | -3.5 (4.7)   | -2.7 (2.5)   | -3.0 (2.2)   | -4.2 (3.3)   | -1.3 (1.2)   |
| ATM |                    | DD     | -3.8 (5.1)   | -3.0 (2.7)   | -3.3 (2.4)   | -4.5 (3.5)   | -1.4 (1.3)   |
|     | DREmax             | Df     | -1.6         | -0.6         | -0.5         | -0.9         | -0.5         |
|     |                    | Dc     | -15.8        | -7.6         | -6.5         | -12.7        | -5.9         |
|     | 11                 | Df     | +0.06 (0.05) | +0.05 (0.04) | +0.04 (0.03) | +0.10 (0.09) | 0+.08 (0.04) |
|     | $\overline{DRE}$   | Dc     | +1.2 (1.5)   | +0.9 (0.6)   | +0.9 (0.6)   | +2.3 (2.1)   | +1.3 (0.9)   |
|     |                    | DD     | +1.3 (1.4)   | +1.0 (2.7)   | +0.9 (0.7)   | +2.4 (0.7)   | +1.4 (1.3)   |
|     | DREmax             | Df     | +0.3         | +0.1         | +0.1         | +0.5         | +0.3         |
| TOA | DKL                | Dc     | +5.5         | +1.9         | +2.6         | +10.0        | +6.4         |
|     |                    | Df     | +0.7 (0.1)   | +0.5 (0.2)   | +0.7 (0.1)   | +1.4 (0.1)   | +1.9 (0.1)   |
|     | DREff              | Dc     | +6.9 (0.2)   | +5.4 (0.9)   | +6.3 (0.4)   | +14.1 (0.7)  | +11.6 (0.9)  |
|     |                    | DD     | +5.1 (0.2)   | +4.0 (0.7)   | +4.6 (0.3)   | +10.3 (0.5)  | +9.0 (0.7)   |
|     | $\delta DRE$       |        | +0.38        | -0.53        | +0.12        | -0.15        | -2.76        |

|         | mean   | +7.7 (4.9) | +4.2 (1.6) | +5.1 (3.6) | +3.1 (2.5) | +12.2 (11.1) |
|---------|--------|------------|------------|------------|------------|--------------|
| ftr_DRE | median | +6.4       | +3.7       | +5.6       | +3.4       | +7.2         |
|         | min    | +0.5       | +2.2       | -3.3       | -5.7       | +0.1         |
|         | max    | +28.6      | +7.6       | +9.9       | +8.5       | +40.7        |

Table 4. The same as Table 3, but for the episode-averaged dust net direct radiative effect (DRE<sub>NET</sub>).

|     | NET              |        | ARN          | GRA          | EVO          | TRJ          | BCN          |
|-----|------------------|--------|--------------|--------------|--------------|--------------|--------------|
|     |                  | Df     | -5.7 (6.6)   | -5.6 (5.0)   | -3.6 (1.4)   | -5.1 (3.7)   | -2.9 (1.8)   |
|     | $\overline{DRE}$ | Dc     | -4.9 (5.2)   | -6.0 (6.5)   | -2.7 (1.4)   | -2.0 (1.8)   | -2.6 (1.5)   |
|     |                  | DD     | -10.6 (11.6) | -11.6 (11.4) | -6.3 (2.4)   | -7.1 (5.3)   | -5.5 (3.1)   |
|     | DREmax           | Df     | -43.0        | -28.0        | -14.7        | -31.3        | -27.1        |
|     | DKE              | Dc     | -50.4        | -41.7        | -20.3        | -36.1        | -33.7        |
|     |                  | Df     | -141.1 (2.1) | -174.6 (6.8) | -131.3 (3.1) | -127.6 (3.1) | -135.5 (5.0) |
| BOA | DREff            | Dc     | -75.7 (1.6)  | -88.6 (6.5)  | -66.6 (2.2)  | -56.3 (2.0)  | -64.1 (3.9)  |
|     |                  | DD     | -94.8 (1.6)  | -113.7 (6.6) | -85.9 (2.4)  | -77.8 (2.2)  | -83.6 (4.1)  |
|     | $\delta DRE$     |        | +0.5         | +0.7         | -0.2         | +5.8         | +2.9         |
|     |                  | mean   | +46.1 (6.0)  | +44.9 (5.1)  | +45.8 (4.5)  | +52.2 (9.6)  | +44.6 (9.9)  |
|     | ftr DRE          | median | +44.7        | +45.5        | +45.8        | +51.5        | +44.5        |
|     | III_DKE          | min    | +60.4        | +52.6        | +55.3        | +67.6        | +84.1        |
|     |                  | max    | +31.6        | +31.4        | +32.9        | +10.5        | +14.5        |
|     | DRE              | Df     | +1.1 (1.8)   | +1.8 (1.2)   | +0.7 (0.4)   | +1.3 (0.9)   | +0.4 (0.4)   |
|     |                  | Dc     | +0.0 (0.0)   | +2.2 (1.8)   | -0.6 (2.2)   | -0.3 (1.4)   | +0.5 (0.8)   |
| ATM |                  | DD     | +1.1 (1.8)   | +4.0 (2.8)   | +0.1 (2.6)   | +1.0 (1.7)   | +0.9 (1.1)   |
|     | DREmax           | Df     | +5.8         | +3.5         | +1.1         | +2.6         | +1.1         |
|     |                  | Dc     | +1.3         | +5.1         | +1.5         | +1.5         | +1.5         |
|     | 10               | Df     | -4.6 (5.0)   | -3.9 (3.9)   | -2.9 (1.5)   | -3.8 (.9)    | -2.4 (1.6)   |
|     | $\overline{DRE}$ | Dc     | -4.9 (5.3)   | -3.8 (5.0)   | -3.3 (5.7)   | -2.3 (2.4)   | -2.2 (1.6)   |
|     |                  | DD     | -9.5 (3.1)   | -7.7 (5.3)   | -6.2 (4.1)   | -6.1 (5.3)   | -4.6 (3.1)   |
|     | DREmax           | Df     | -30.1        | -20.4        | -17.9        | -23.2        | -22.8        |
|     | DKE              | Dc     | -49.3        | -26.0        | -27.1        | -32.2        | -27.8        |
| TOA |                  | Df     | -104.9 (3.4) | -122.7 (9.5) | -101.6 (6.1) | -91.3 (4.3)  | -112.3 (4.0) |
| IUA | DREff            | Dc     | -54.3 (2.8)  | -55.5 (8.0)  | -52.6 (5.0)  | -37.2 (3.5)  | -46.5 (3.3)  |
|     |                  | DD     | -69.0 (2.9)  | -75.2 (8.3)  | -67.2 (5.2)  | -53.5 (3.7)  | -64.6 (3.4)  |
|     | δDRE             |        | +0.3         | +0.4         | +0.4         | +5.8         | +2.9         |
|     |                  | mean   | +48.8 (7.7)  | +55.7 (11.6) | +55.9 (15.2) | +57.7 (16.3) | +52.6 (15.3) |
|     | ftr_DRE          | median | +47.3        | +53.6        | +52.0        | +53.1        | +50.4        |
|     |                  | max    | +65.9        | +74.5        | +94.4        | +97.9        | +94.9        |
|     |                  | •      |              |              |              |              |              |

min +30.9 +42.4 +34.2 +39.0 +15.6

Table 5. Mean (std), maximal (Max) and minimal (Min) values together the percentiles P(25), P(50) and P(75) of  $\Delta DRE_{LW}$  (W m<sup>-2</sup>) and  $\Delta^{rel}DRE_{LW}$  (%) at BOA and TOA, std stands for the standard deviation.

|                                  |                            | Mean         | Min   | Max   | P(25) | P(50) | P(75) |
|----------------------------------|----------------------------|--------------|-------|-------|-------|-------|-------|
|                                  |                            |              | TOA   |       |       |       |       |
|                                  | All dataset                | +0.1 (0.5)   | -1.2  | +2.6  | -0.06 | +0.01 | +0.1  |
| $\Delta DRE_{LW}$                | Fine $r_g 

Table 6. The same as Table 5, but for  $\Delta DRE_{NET}$  (W m<sup>-2</sup>) and  $\Delta^{rel}DRE_{NET}$  (%).

|                                  |                                                  | Mean        | Min   | Max   | P(25) | P(50) | P(75) |
|----------------------------------|--------------------------------------------------|-------------|-------|-------|-------|-------|-------|
|                                  |                                                  |             | TOA   |       |       |       |       |
|                                  | All dataset                                      | -0.5 (2.6)  | -14.4 | +2.3  | -0.3  | +0.3  | +0.7  |
| ADDE                             | SZA < 70°                                        | +0.2 (1.4)  | -10.4 | +2.3  | +0.1  | +0.5  | +0.8  |
| $\Delta DRE_{NET}$               | $\mathrm{DRE^{(II)}} \leq -20~\mathrm{W~m^{-2}}$ | -2.0 (2.6)  | -10.4 | +2.3  | -3.1  | -1.8  | -0.3  |
|                                  | $DRE^{\rm (II)}\!>\!\text{-}20~W~m^{\text{-}2}$  | +0.5 (0.5)  | -1.2  | +2.3  | +0.2  | +0.5  | +0.8  |
| $\Delta^{\mathrm{rel}}DRE_{NET}$ | All dataset                                      | -3.5 (17.6) | -79.5 | +66.3 | -11.9 | -5.1  | +2.4  |
| Δ DRENET                         | SZA < 70°                                        | -8.5 (13.0) | -79.5 | +15.8 | -13.4 | -7.0  | -1.2  |
|                                  |                                                  |             | BOA   |       |       |       |       |
|                                  | All dataset                                      | -0.06 (2.4) | -14.4 | +6.4  | -0.1  | +0.4  | +0.9  |
| ADDE                             | $SZA < 70^{\circ}$                               | +0.5 (1.5)  | -6.4  | +6.4  | +0.2  | +0.5  | +1.1  |
| $\Delta DRE_{NET}$               | $\mathrm{DRE^{(II)}} \leq -20~\mathrm{W~m^{-2}}$ | +0.5 (2.5)  | -4.9  | +6.4  | -1.1  | +0.8  | +1.9  |
|                                  | $DRE^{\rm (II)}\!>\!-20~W~m^{\text{-}2}$         | +0.6 (0.8)  | -6.4  | +3.1  | +0.3  | +0.5  | +0.9  |
| $\Delta^{\mathrm{rel}}DRE_{NET}$ | All dataset                                      | -1.1 (14.0) | -34.2 | +92.7 | -8.2  | -4.0  | +1.0  |
| $\Delta^{res}DRE_{NET}$          | $SZA 

Table 7. Maxima of the hourly aerosol heating rates (AHR<sub>i</sub><sup>max</sup>, K day<sup>-1</sup>) found for the entire episode at each lidar stations. Values for the fine (Dc), coarse (Dc) and total dust (DD) are shown. The episode-averaged of those AHR<sub>i</sub><sup>max</sup> (AHR<sub>i</sub>, K day<sup>-1</sup>) and their corresponding heights (z\_AHR<sub>i</sub>, km), and the fine-to-total AHR ratio (ftr\_AHR<sub>i</sub>) is also shown. The index i stands for the SW, LW and NET ranges. The standard deviation (std) is shown in brackets.

|                            |    | ARN                         | GRA                         | EVO                         | TRJ                         | BCN                         |
|----------------------------|----|-----------------------------|-----------------------------|-----------------------------|-----------------------------|-----------------------------|
|                            | Df | +0.75                       | +0.25                       | +0.24                       | +0.55                       | +0.21                       |
| $AHR_{SW}^{max}$           | Dc | +1.96                       | +0.71                       | +0.54                       | +1.23                       | +0.48                       |
|                            | DD | +2.71                       | +0.96                       | +0.78                       | +1.78                       | +0.69                       |
|                            | Df | 3.2 (3.1)                   | 5.0 (3.5)                   | 3.6 (3.3)                   | 3.7 (3.3)                   | 1.4 (1.3)                   |
| $\overline{z}_AHR_{SW}$    | Dc | 3.6 (3.4)                   | 5.8 (3.7)                   | 4.3 (3.5)                   | 4.4 (3.7)                   | 2.6 (3.2)                   |
|                            | DD | 3.8 (3.5)                   | 5.3 (3.9)                   | 3.7 (3.6)                   | 4.3 (3.7)                   | 2.0 (2.6)                   |
|                            | Df | +0.11 (0.16)                | +0.12 (0.06)                | +0.06 (0.07)                | +0.14 (0.13)                | +0.05 (0.06)                |
| $\overline{AHR_{SW}}$      | Dc | +0.26 (0.36)                | +0.30 (0.18)                | +0.21 (0.12)                | +0.31 (0.30)                | +0.16 (0.12)                |
|                            | DD | +0.37 (0.30)                | +0.42 (0.20)                | +0.27 (0.18)                | +0.45 (0.43)                | +0.21 (0.17)                |
| ftr_AHRsw                  |    | +30 (7)                     | +29 (4)                     | +29 (5)                     | +33 (10)                    | +28 (6)                     |
|                            | Df | -0.05                       | -0.01                       | -0.01                       | -0.06                       | -0.01                       |
| $AHR_{LW}^{max}$           | Dc | -0.53                       | -0.11                       | -0.17                       | -0.79                       | -0.15                       |
|                            | DD | -0.58                       | -0.12                       | -0.18                       | -0.85                       | -0.16                       |
|                            | Df | 1.0 (0.2)                   | 1.4 (0.9)                   | 1.2 (1.1)                   | 1.4 (1.0)                   | 1.7 (0.8)                   |
| $\overline{z_{-}AHR_{LW}}$ | Dc | 1.1 (0.3)                   | 1.2 (0.4)                   | 1.2 (0.8)                   | 1.6 (1.0)                   | 1.7 (1.0)                   |
|                            | DD | 1.1 (0.3)                   | 1.2 (0.4)                   | 1.2 (0.8)                   | 1.6 (1.0)                   | 1.7 (0.8)                   |
|                            | Df | -4.2 (9.9) 10 <sup>-3</sup> | -3.5 (3.6) 10 <sup>-3</sup> | -3.5 (3.9) 10 <sup>-3</sup> | -8.7 (8.5) 10 <sup>-3</sup> | -4.1 (3.7) 10 <sup>-3</sup> |
| $\overline{AHR_{LW}}$      | Dc | -0.05 (0.11)                | -0.04 (0.05)                | -0.04 (0.03)                | -0.12 (0.20)                | -0.04 (0.04)                |
|                            | DD | -0.05 (0.11)                | -0.04 (0.05)                | -0.04 (0.03)                | -0.12 (0.20)                | -0.04 (0.04)                |
| ftr_AHR <sub>LW</sub>      |    | +16 (19)                    | +12 (11)                    | +11 (15)                    | +6 (8)                      | +15 (16)                    |
|                            | Df | +0.60                       | +0.25                       | +0.20                       | +0.54                       | +0.21                       |
| $AHR_{NET}^{max}$          | Dc | +1.23                       | +0.69                       | +0.39                       | +1.18                       | +0.48                       |
|                            | DD | +1.83                       | +0.94                       | +0.59                       | +1.72                       | +0.69                       |
|                            | Df | 3.3 (1.3)                   | 3.7 (1.3)                   | 3.0 (1.7)                   | 3.2 (2.0)                   | 5.0 (1.6)                   |
| $z\_AHR_{NET}$             | Dc | 3.4 (1.3)                   | 3.7 (1.3)                   | 3.0 (1.7)                   | 3.2 (2.0)                   | 4.9 (1.6)                   |
|                            | DD | 3.4 (1.3)                   | 3.7 (1.3)                   | 3.0 (1.7)                   | 3.2 (2.0)                   | 4.9 (1.7)                   |
| $\overline{AHR_{NET}}$     | Df | +0.08 (0.11)                | +0.11 (0.07)                | +0.10 (0.04)                | +0.14 (0.14)                | +0.06 (0.05)                |
| An K <sub>NET</sub>        | Dc | +0.18 (0.26)                | +0.27 (0.17)                | +0.23 (0.09)                | +0.30 (0.30)                | +0.16 (0.12)                |

| DD                     | +0.26 (0.37) | +0.38 (0.20) | +0.33 (0.10) | +0.44 (0.31) | +0.22 (0.14) |
|------------------------|--------------|--------------|--------------|--------------|--------------|
| ftr_AHR <sub>NET</sub> | +32 (6)      | +29 (4)      | +31 (3)      | +34 (11)     | +29 (7)      |