# Peer review of "Fine and coarse dust radiative impact during an intense Saharan dust outbreak over the Iberian Peninsula - long-wave and net direct radiative effect"

_EGUsphere, 2025_

## Author Comment (AC2)

The authors highly appreciate the thorough review provided by all the reviewers that has significantly improved the quality of the manuscript. For clarity, the revisions addressing the comments from Reviewer #1 (R1) and Reviewer #2 (R2) are highlighted in purple and blue, respectively, and included in one single file because of common comments. Changes that respond to comments from both reviewers are marked in red. In addition, Figure 1 has been modified. The Abstract and the rest of manuscript has also been modified accordingly with the changes introduced in the manuscript, including the final conclusions has been substantially revised by accounting for all the reviewers' suggestions, comments and recommendations.

New Figures and Tables in the Supplementary Material have been included, renumbering thus the rest of Figures and Tables.

Changes referred to given pages and lines correspond to those of the revised version of the manuscript.
* * *
**Reviewer #1's comments:**

This manuscript presents a comprehensive analysis of long-wave (LW) and net direct radiative effects (DRE) during an intense Saharan dust outbreak over the Iberian Peninsula in March-April 2021. The study employs lidar observations from five stations combined with the GAME radiative transfer model to quantify the separate contributions of fine (Df) and coarse (Dc) dust particles to dust LW DRE. This work complements the authors' previous study on short-wave (SW) effects (López-Cayuela et al., 2025) and introduces a novel comparison between two methodological approaches: (1) calculating DRE by separating Df and Dc contributions versus (2) treating total dust as a single component. The science merits publication in ACP, but major revisions are needed for the following concerns.

**Reviewer's comment (R1C1):** The paper need to discuss that why the estimation of dust DRE by simulating the contribution of Df and Dc components separately (the major contribution of this study) is a better option (or is necessary) compared to directly simulating DRE for the total dust component as a whole. For example, what would be the advantage by doing this? Or what is the improvement by doing this? The motivation should be clearly stated in the introduction section.

**Authors' response**: Authors thank the reviewer for the suggestion. To highlight this question on the separation of dust (fine and coarse) modes for improving the radiative effect estimation, the following paragraph has been added into the manuscript:

Page 2, lines 66-82 "Recent research has demonstrated that radiative transfer models must handle fine and coarse modes separately to accurately represent the radiative effects of mineral dust. Sicard et al. (2014a) found that a clear distinction between dust modes is critical for the reliable estimation of longwave radiative forcing, particularly in the presence of large particles, which are common during mineral dust outbreaks, and mostly with intense dust incidence. This requirement is further supported by Adebiyi et al. (2020), who discovered that the atmosphere burden of coarse dust is approximately four times larger than that simulated by current climate models. Consequently, an inadequate representation of coarse particles can lead to substantial errors in modeled dust–climate interactions. The findings also highlight the contrasting radiative effects associated with the two size modes, with coarse dust inducing a net warming at the top of the atmosphere and fine dust contributing to cooling it, indicating their fundamentally different roles in the Earth's radiative budget. In addition, Kok et al. (2017) demonstrated that the dust direct radiative effect is highly sensitive to the atmospheric dust size distribution, with climate models systematically underestimating the coarse-mode dust while overestimating the fine-mode fraction. Their analytical framework highlights the importance of considering the size-resolved dust mass and distribution, given that key radiative properties of dust (such as singlescattering albedo, asymmetry parameter, and extinction efficiency) are strongly dependent on particle size.

When considered collectively, these studies demonstrate that fine and coarse dust must not be treated as a single, homogeneous aerosol population if radiative accuracy is to be preserved. The two modes differ not only in abundance and lifetime but also in their optical characteristics and radiative impacts. Thus, explicitly separating fine and coarse modes in radiative transfer models is crucial to reduce persistent biases in estimates of dust radiative forcing."

The following references have been also included:

- Sicard, M., Bertolín, S., Muñoz, C., Rodríguez, A., Rocadenbosch, F., and Comerón, A. (2014). Separation of aerosol fine-and coarse-mode radiative properties: Effect on the mineral dust longwave, direct radiative forcing. *Geophysical research letters*, 41 (19), 6978-6985.
- Adebiyi, A. A., and Kok, J. F. (2020). Climate models miss most of the coarse dust in the atmosphere. *Science advances*, 6 (15), eaaz9507.
- Kok, J. F., Ridley, D. A., Zhou, Q., Miller, R. L., Zhao, C., Heald, C. L., ... and Haustein, K. (2017). Smaller desert dust cooling effect estimated from analysis of dust size and abundance. *Nature Geoscience*, 10 (4), 274-278.

**Reviewer's comment (R1C2):** Assuming night-time hourly LW fluxes to be equal to the mean value of the daytime LW ones may not be appropriate as there could be strong diurnal variation of dust AOD overland (Tindan et al., 2023; Tindan et al., 2025), so as the LST (as shown in Figure 1a). A justification should be made by either providing evidence of insignificant diurnal variation of dust LW DRE in previous studies, or modifications in calculating night-time hourly LW fluxes separately from day-time hourly LW fluxes.

**Authors' response**: Authors thank the reviewers for their comment. The authors agree that residual uncertainty may persist. However, the previous studies of Tindan et al. (2023, 2025) indicate that diurnal dust variability over the Iberian Peninsula is negligible. In addition, the moderate daily variability of the downward long-wave flux (approximately 13%, see Granados-Muñoz et al., 2019) falls within the uncertainty range of the radiative forcing estimates. Therefore, using the daytime means of longwave flux to represent nighttime conditions does not significantly affect the results. Thus, the following text has been added to the manuscript to justify the assumption made for day- and night-time LW fluxes:

Page 5, lines 187-195: "Under this assumption, some uncertainty may still arise from two main factors affecting the DRE: differences in DOD values, and variations in the downward radiation flux during both day- and night-time. Regarding the first factor, the episode analyzed was highly cloudy, resulting in numerous observational gaps during both day and night. However, recent studies by Tindan et al. (2023, 2025) investigating diurnal differences in dust aerosols across the dust belt have shown that such variations are insignificant over the Iberian Peninsula. With respect to the second factor, Granados-Muñoz et al. (2019) reported that the downward longwave radiation flux displays a moderate daily variability (approximately 13%), which would slightly modify the contrast in radiative forcing between daytime and nighttime conditions. However, this variability falls within the uncertainty range of the radiative forcing difference. Consequently, assuming night-time hourly LW fluxes to be equal to the mean daytime LW flux does not significantly affect the results of the present study."

The following references have been also added to the reference list:

- Tindan, J. Z., Jin, Q., and Pu, B.: Understanding day–night differences in dust aerosols over the dust belt of North Africa, the Middle East, and Asia, Atmos. Chem. Phys., 23, 5435-5466, 10.5194/acp-23-5435-2023, 2023.
- Tindan, J. Z., Pu, B., and Jin, Q.: Trends in daytime and nighttime dust aerosols over the Dust Belt revealed by IASI, Science of The Total Environment, 1004, 180742, https://doi.org/10.1016/j.scitotenv.2025.180742, 2025.

**Reviewer's comment (R1C3):** The extinction coefficient of coarse portion of dust be larger than the extinction of total dust in LW, is not physically meaningful, as fine particles has non-negative extinction coefficient in LW. From my understanding, α_LW / α_532 ratio for coarse mode is coarse-mode α_LW / coarse-mode α_532, while total α_LW / α_532 ratio is total α_LW / total α_

Consider total α_532 > coarse-mode α_532, and total α_LW could be similar to total α_LW due to low sensitivity of fine mode in LW, it could therefore lead to greater coarse-mode α_LW / α_532 ratio than total α_LW / α_532 ratio, as shown in Figure 1c. However, it does not mean coarse-mode α_LW > total α_LW. Therefore, I suggest the authors clarify such statement in the manuscript (lines 223-226; lines 533-536). It is also suggested to plot the absolute value of α_532 and the Mie-calculated α_LW (total, fine and coarse) in the LW spectrum, which is helpful for the above explanations.

**Authors' response**: Authors strongly appreciate this reviewer's comment, and this is clarified by adding the following text:

Page 7, lines 264-268: "It should be mentioned that values for $\alpha_{\delta LW}^{coarse}/\alpha_{532}^{coarse}$ are slightly higher than those for $\alpha_{\delta LW}^{total}/\alpha_{532}^{total}$. Those ratios depicted in Figure 1c are normalized against the corresponding $\alpha_{532}$ for coarse (Dc) and total dust, respectively. Thus, the $\alpha_{LW}^{coarse}/\alpha_{532}^{coarse}$ ratio is slightly higher due to a slightly smaller $\alpha_{532}^{coarse}$ value as compared to $\alpha_{532}^{total}$. Furthermore, due to the low sensitivity of fine mode in LW, $\alpha_{\delta LW}^{fine}/\alpha_{532}^{fine}$, is an order of magnitude lower than $\alpha_{\delta LW}^{coarse}/\alpha_{532}^{coarse}$, with mean differences in the ratio of less than 6% accounting for all the lidar stations on average along the dust episode."

In addition, a new equation (Eq. 3) has been added (see also the response to R1C4 comment), renumbering all the following equations, and the Figure 1 has been modified and replaced by:

[Figure]

**Figure 1. (a)** Hourly land surface temperature (LST, in °C), where the red dots represent the cases coincident with lidar measurements; AERONET geometric median radius ($r_g$, in μm ) and standard deviation ($\sigma_g$) for the **(b)** fine and **(c)** coarse modes, where the dashed lines represent the linear fitting of $r_g$ over time; Episode-averaged values of **(d)** the Mie-derived normalized spectral extinction ($\alpha_{LW}(Mie)/\alpha_{532}(Mie)$) (see Eq. 3), **(e)** asymmetry factor ($g_{LW}$), and **(f)** single scattering albedo ($\omega_{LW}$), for the fine (blue), coarse (red) and total (yellow) modes. All the panels refer to El Arenosillo/Huelva (ARN) station; for the rest of stations, see the Supplementary Material.

**Reviewer's comment (R1C4):** Another question is, how does the authors calculate α_LW in Df and Dc modes? For example, in Dc mode, are they calculated as α_LW,Dc = α_532 * α_LW,Dc / α_532,Dc? or as α_LW,Dc = α_532,Dc * α_LW,Dc / α_532, Dc? I suggest providing the equations of calculating α_LW in the manuscript.

**Authors' response**: Following the authors' response to R1C3, the Df and Dc simulations are run separately. Therefore, $\alpha_{LW}^{Dc}$ is normalized by the corresponding $\alpha_{532}^{Dc}$, and $\alpha_{LW}^{Df}$ is normalized by the corresponding $\alpha_{532}^{Df}$. For clarifying, the following text and a new equation (Eq. (3)) have been added in the manuscript:

Pages 4-5, lines 168-172: "The Mie module is capable of computing the spectral single scattering albedo ($\omega_{LW}$), the asymmetry factor ($g_{LW}$) and the normalized extinction coefficient ($\alpha_{LW}/\alpha_{532}$) for each atmospheric layer. Then, the estimated extinction coefficient in the LW range, $\alpha_{LW}^i(estimated)$, distinguishing between Df and Dc modes, is calculated as follows:

$$\alpha_{LW}^i(estimated) = \alpha_{532}^i(measured) \times \left. \alpha_{LW}^i(Mie) \middle/ \alpha_{532}^i(Mie) \right., \qquad (3)$$

where the upper-index $i$ refers to total dust, Dc and Df, and $\alpha_{532}(measured)$ is the extinction coefficient at 532 nm as obtained in López-Cayuela et al. (2023)."

Minor concerns:

**Reviewer Comment (R1C5)**: Line 536: "These findings align with previous literature and were validated in Section 4." Section 4 is summary and conclusion, there is no validation in Section 4. Please clarify.

**Authors' response**: Thank you for pointing out the wrong section. It has been corrected in the new version of the manuscript: "These findings align with previous literature and were validated in Section 3.1".
* * *
**Reviewer #2's comments:**

This study estimates the temporal variation of direct radiative effects (DRE) of fine and coarse dust in the LW and net DRE during an intense dust outbreak over the Iberian Peninsula using lidar observations from five stations and the GAME radiative transfer model. The differences in dust DRE obtained from the combination of separately computed coarse and fine dust contributions versus the classical approach (no separation) are examined. The aerosol heating rate is also discussed. The analyses are comprehensive, but major revisions are needed to address several concerns and to present the results more clearly and concisely.

**Reviewer's comment (R2C1):** Given the absence of flux measurements for evaluation, it is unclear if separating dust into fine and coarse modes and computing their properties independently yields more accurate results than the classical approach. A clear conceptual justification is needed.

**Authors' response**: This comment is aligned with the response to the reviewer #1's comment (R1C1). Authors thank the reviewer for the suggestion. The following paragraph has been added into the manuscript:

Page 2, lines 66-82: "Recent research has demonstrated that radiative transfer models must handle fine and coarse modes separately to accurately represent the radiative effects of mineral dust. Sicard et al. (2014) found that a clear distinction between dust modes is critical for the reliable estimation of longwave radiative forcing, particularly in the presence of large particles, which are common during mineral dust outbreaks, and mostly with intense dust incidence. This requirement is further supported by Adebiyi et al. (2020), who discovered that the atmosphere burden of coarse dust is approximately four times larger than that simulated by current climate models. Consequently, an inadequate representation of coarse particles can lead to substantial errors in modeled dust–climate interactions. The findings also highlight the contrasting radiative effects associated with the two size modes, with coarse dust inducing a net warming at the top of the atmosphere and fine dust contributing to cooling it, indicating their fundamentally different roles in the Earth's radiative budget. In addition, Kok et al. (2017) demonstrated that the dust direct radiative effect is highly sensitive to the atmospheric dust size distribution, with climate models systematically underestimating the coarse-mode dust while overestimating the fine-mode fraction. Their analytical framework highlights the importance of considering the size-resolved dust mass and distribution, given that key radiative properties of dust (such as single-scattering albedo, asymmetry parameter, and extinction efficiency) are strongly dependent on particle size.
When considered collectively, these studies demonstrate that fine and coarse dust must not be treated as a single, homogeneous aerosol population if radiative accuracy is to be preserved. The two modes differ not only in abundance and lifetime but also in their optical characteristics and radiative impacts. Thus, explicitly separating fine and coarse modes in radiative transfer models is crucial to reduce persistent biases in estimates of dust radiative forcing."

The following references have also added to the manuscript:
- Sicard, M., Bertolín, S., Muñoz, C., Rodríguez, A., Rocadenbosch, F., and Comerón, A. (2014). Separation of aerosol fine-and coarse-mode radiative properties: Effect on the mineral dust longwave, direct radiative forcing. *Geophysical research letters*, *41*(19), 6978-6985.
- Adebiyi, A. A., and Kok, J. F. (2020). Climate models miss most of the coarse dust in the atmosphere. *Science advances*, *6* (15), eaaz9507.
- Kok, J. F., Ridley, D. A., Zhou, Q., Miller, R. L., Zhao, C., Heald, C. L., ... and Haustein, K. (2017). Smaller desert dust cooling effect estimated from analysis of dust size and abundance. *Nature Geoscience*, *10* (4), 274-278.

**Reviewer's comment (R2C2):** The uncertainties of DRE should be estimated, or at least the main uncertainty sources should be discussed. It is possible that the uncertainty magnitude exceeds the difference between the two approaches. Consider uncertainties from lidar measurements, AERONET data, the GAME model, etc.

**Authors' response**:
The authors consider that the uncertainties in DRE are difficult to quantify, since they are not only associated with uncertainties in the input parameters to the radiative transfer model used (GAME), but also with the different assumptions made by GAME. A sensitivity study varying all input parameters is beyond the scope of this article. In addition, we wanted to follow the same procedure that we applied in our previous study of the DRE in the SW range (López-Cayuela et al., 2025). In the analysis of differences, we have assumed that all input parameters are fixed (see Table 1). Then, the differences in DRE as estimated between separating or not separating the dust components are purely examined. Nevertheless, it is true that this issue deserves to be highlighted, and then the following paragraphs have been included to address this matter, providing an overview of the main sources of uncertainty in the main input parameters introduced in GAME:

Page 3, lines 118-120: "The uncertainties in the $\alpha^{532}$ calculation by using this method are 30–50 %, 20–30 %, and 15–25 % for Df, Dc, and DD (=Dc+Df) dust, respectively (Ansmann et al., 2019)."
Page 4, lines 137-138: "Particularly, the hourly LST V2 dataset is used, which has uncertainties of less than 0.5%."
Page 4, lines 151-158: "It should be highlighted that the refractive index used in the simulations (Volz, 1983), although assumed no varying, could be a source of uncertainty. Di Biagio et al. (2014, 2017) investigated the variability of the refractive index of mineral dust in LW as a function of its mineralogical composition and size distribution using in situ measurements. That study suggested that while a constant real refractive index can be probably assumed for dust from different sources, a varying complex refractive index should be used both at global and regional scale. They reported that for Saharan dust sampled at various sites, the real refractive index ranged from 1.3 to 2.0, and the complex refractive index ranged from 0.3 to 0.9 at a wavelength of 10 µm. The refractive index reported by Volz (1983), which has been used in the GAME simulations, is within those intervals of values for both the real and complex refractive index."

In addition, the following references have included:
- Di Biagio, C., Boucher, H., Caquineau, S., Chevaillier, S., Cuesta, J., and Formenti, P.: Variability of the infrared complex re fractive index of African mineral dust: experimental estimation and implications for radiative transfer and satellite remote

sens ing, Atmos. Chem. Phys., 14, 11093-11116, https://doi.org/10.5194/acp-14-11093-2014, 2014.

- Di Biagio, C., Formenti, P., Balkanski, Y., Caponi, L., Cazaunau, M., Pangui, E., Journet, E., Nowak, S., Caquineau, S., Andreae, M. O., Kandler, K., Saeed, T., Piketh, S., Seibert, D., Williams, E., and Doussin, J.-F.: Global scale variability of the mineral dust long-wave refractive index: a new dataset of in situ measurements for climate modeling and remote sensing, Atmos. Chem. Phys., 17, 1901–1929, https://doi.org/10.5194/acp-17-1901-2017, 2017.

Specific Issues:

**Reviewer's comment (R2C3):** Page 1, line 25: Change "and hence the atmospheric DREnet was positive" to "and the derived atmospheric DREnet was positive" given the unclear causal relationship in the sentence.

**Authors' response:** The authors thank the reviewer for the suggestion, and the sentence has been modified in accordance with the proposed change.

**Reviewer's comment (R2C4):** Page 1, line 30: Be cautious when using "underestimate" or "overestimate," as there are no reference data to determine accuracy.

**Authors' response:** The corresponding paragraph has been modified as follows:

Page 1, lines 29-34: "As a novelty of this study, two methodologies for estimating DRE in both LW and net spectral ranges are compared. Differences in DRE between a classical approach considering total dust and an approach separating fine and coarse modes are analysed. $DRE_{LW}$ (and $DRE_{NET}$) is apparently underestimated (overestimated) by using the dust-mode separation approach in comparison to the classical one (no separation) when fine radii are lesser (greater) than a particular threshold (e.g., 0.1 µm), revealing the particle size impact in $DRE_{LW}$".

**Reviewer's comment (R2C5):** Page 2, line 65: Station abbreviations, locations, and the dust outbreak period should be mentioned in the methods section, not in the introduction. A site map in the Supplement would also be helpful, given the frequent discussion of site-to-site differences.

**Authors' response:** The authors agree with the reviewer. This information have been moved to Section 2, adding a new section: '2.1 Monitoring stations and lidar measurements' (summarizing what is already reported in López-Cayuela et al., 2023), renumbering thus the following sections of the first version of the manuscript. Modifications have been performed in the Introduction (page 3, lines 102-120). Moreover, a map showing the locations of the stations has also been added in the Supplement Material (Figure S1), and new references have been added.

[Figure]

**Figure S1. MODIS image of the corrected reflectance over the Iberian Peninsula on 31 March 2021. The five Iberian lidar stations are marked with a red dot (from North-East to South-West in the Iberian Peninsula): Barcelona (BCN), Torrejón/Madrid (TRJ), Évora (EVO), Granada (GRA), and El Arenosillo/Huelva (ARN) sites.**

**Reviewer's comment (R2C6):** Page 2, line 75: Specify which components are meant in "separation of both components".

**Authors' response:** The original sentence has been changed to:

Page 3, lines 88-89: "… only a few studies have addressed the separation of both fin dust (Df) and coarse dust (Dc) components …"

**Reviewer's comment (R2C7):** Page 3, line 105: Avoid unnecessary acronyms throughout the paper, e.g., those for surface albedo here and for supplementary materials elsewhere, since the paper already includes numerous variable acronyms.

**Authors' response:** Agree. The manuscript has been modified in accordance with the proposed change. In particular, the acronyms referred to Iberian Peninsula, radiative forcing, surface albedo and Supplementary Material have been removed in the manuscript.

**Reviewer's comment (R2C8):** Page 3, line 110: Suggest comparing this refractive index data with more recent datasets, such as Di Biagio et al., 2017 (Atmospheric Chemistry and Physics, 17, 1901–1929). This is also an important source of uncertainty.

**Authors' response:** The following paragraph has been added to the manuscript, following the reviewer's suggestion:

Page 4, lines 151-158: "It should be highlighted that the refractive index used in the simulations (Volz, 1983), although assumed no varying, could be a source of uncertainty. Di Biagio et al. (2014, 2017) investigated the variability of the refractive index of mineral dust in LW as a function of its mineralogical composition and size distribution using in situ measurements. That study suggested that while a constant real refractive index can be probably assumed for dust from different sources, a varying complex refractive index should be used both at global and regional scale. They reported that for Saharan dust sampled at various sites, the real refractive index ranged from 1.3 to 2.0, and the complex refractive index ranged from 0.3 to 0.9 at a wavelength of 10 μm. The refractive index reported by Volz (1983), which has been used in the

GAME simulations, is within those intervals of values for both the real and complex refractive index."

In addition, the following references has been also added:
- Di Biagio, C., Boucher, H., Caquineau, S., Chevaillier, S., Cuesta, J., and Formenti, P.: Variability of the infrared complex re fractive index of African mineral dust: experimental estimation and implications for radiative transfer and satellite remote sens ing, Atmos. Chem. Phys., 14, 11093-11116, https://doi.org/10.5194/acp-14-11093-2014, 2014.
- Di Biagio, C., Formenti, P., Balkanski, Y., Caponi, L., Cazaunau, M., Pangui, E., Journet, E., Nowak, S., Caquineau, S., Andreae, M. O., Kandler, K., Saeed, T., Piketh, S., Seibert, D., Williams, E., and Doussin, J.-F.: Global scale variability of the mineral dust long-wave refractive index: a new dataset of in situ measurements for climate modeling and remote sensing, Atmos. Chem. Phys., 17, 1901–1929, https://doi.org/10.5194/acp-17-1901-2017, 2017.

**Reviewer's comment (R2C9):** Page 5, line 200: Clarify what "those values" refer to.

**Authors' response:** The sentence refers to $\gamma$, which is the slope of the linear fitting of $r_g$ over time. The change has been introduced in the revised version of the manuscript as follows:

Page 7, lines 249-251: "Similarly to the fine mode, that increase/decrease over time was not significant either (lower than 2% μm day$^{-1}$) except for BCN, reaching almost 7% μm day$^{-1}$.".

**Reviewer's comment (R2C10):** Page 7, line 250: Remind readers what δDRE represents.

**Authors' response:** The corresponding sentence has been modified as follows:

Page 8, lines 300-301: "The slope of the linear fitting of $DRE_{LW}$ over time ($\delta DRE_{LW}$) showed values that ranged from -2.57 (BCN) to +0.65 (EVO) % day$^{-1}$ (Table 3)."

**Reviewer's comment (R2C11):** Page 8, line 295: Several connecting words (e.g., indeed, moreover, being) are used unnecessarily or incorrectly in multiple places. Suggest revising for smoother flow.

**Authors' response:** The authors wish to thank the reviewer for the suggestion, and the overall manuscript has been revised, including the English spelling and grammar, and modified in accordance with the proposed revision.

**Reviewer's comment (R2C12):** Page 8, line 310: Correct "being 45% in and the SW range"; the word "being" is misused here and elsewhere; revise accordingly.

**Authors' response:** The authors wish to thank the reviewer for the suggestion, and this sentence has been modified:

Page 9, lines 364-365: "In this work, ftr_DRE is less than 12% in the LW range (Sect. 3.2), and 45% in the SW range (López-Cayuela et al., 2025).", in addition to where needed elsewhere throughout the manuscript.

**Reviewer's comment (R2C13):** Page 8, 315: Similarly, suggest modifying "According to other studies, similar results are found, being the DRELW/DRESW greater at BOA than at TOA" to

"Other studies also report higher DRELW/DRESW values at BOA than at TOA" to avoid misuse of "being" and reduce redundancy. Redundant expressions should also be avoided throughout the manuscript to improve readability and conciseness.

**Authors' response:** Authors thank the reviewer for the suggestion. The manuscript has been revised, modifying accordingly the sentences.

**Reviewer's comment (R2C14):** Page 9, line 365: Suggest changing "impact" to "temporal change" for clarity.

**Authors' response:** Thanks. This term has been changed.

**Reviewer's comment (R2C15):** Page 9, line 375: The maximum hourly values at ATM in Figure 4 appear to be much higher than +5.8 (+1.3) W m-2; please verify.

**Authors' response:** Authors thank to the reviewer for pointing out this mistyping. The paragraph has been corrected as follows:

Page 11, lines 430-433: "The maximum hourly $DRE_{NET}$ values at ATM are found at ARN and TRJ, reaching +14.0 (+19.4) W $m^{-2}$ and +11.6 (+19.2) W $m^{-2}$ for Df (Dc) particles, respectively. At the remaining stations, the maximum hourly $DRE_{NET}$ values at ATM ranged from +5.0 to +8.4 W $m^{-2}$ for Df particles and from +8.3 to +9.2 W $m^{-2}$ for Dc particles (see Table 4)."

**Reviewer's comment (R2C16):** Page 10, line 385: Provide context for why SZA < 70 is specified here.

**Authors' response:** The text has been properly changed.

Page 11, lines 440-446: "As discussed in López-Cayuela et al. (2025), the significant $\Delta^{rel}DRE_{SW}$ values found for SZA > 70° are attributed to the intrinsic uncertainty in GAME simulations arising from the assumption of a plane-parallel atmosphere, and, hence, these values should be discarded. However, no clear correlation was observed between $\Delta^{rel}DRE_{LW}$ and SZA. At BOA (TOA), mean $\Delta^{rel}DRE_{LW}$ values of approximately +8.5% (+6.5%) were obtained, although relatively large standard deviations were observed (~25-27%, see Table 5). Indeed, comparable $\Delta^{rel}DRE_{LW}$ values are found for SZA < 70° (see Table 5). Moreover, no clear relationship is evident between $\Delta^{rel}DRE_{LW}$ and DD DOD[532]."

**Reviewer's comment (R2C17):** Page 10, line 405: Use "underestimation" and "overestimation" to describe the traditional approach, not the dust-mode separation approach, if the latter is considered more accurate.

**Authors' response:** Authors understand the reviewer's point of view. However, we aim to highlight that, when using total dust DRE as a reference (i.e., the classical approach), the results obtained from the dust component separation either overestimate or underestimate it. This criterion was the same employed in López-Cayuela et al. (2025). To clarify these aspects, the following sentence has been added to the manuscript as follows:

Page 5, lines 205-207: "As in López-Cayuela et al. (2025), the classical approach (i.e., without dust component separation) is adopted as the reference. Accordingly, throughout this manuscript, cases are described in which the component-separated DRE either overestimates or underestimates this classical approximation."

**Reviewer's comment (R2C18):** Page 10, line 410: The lower mean values for finer rg >= 0.1 um seem inconsistent with the earlier discussion; please clarify.

**Authors' response:** Thank you for pointing out the mistyping. The sentences are corrected as follows:

Pages 11, lines 467-470: "In terms of mean values, the largest differences are found for size distributions dominated by finer particles, for which $\Delta DRE_{LW}$ exhibited mean (std) values of -0.04 (0.58) and -0.03 (0.22) W m$^{-2}$ at BOA and TOA, respectively. In contrast, for cases with fine $r_g \geq$ 0.1 μm, $\Delta DRE_{LW}$ presented mean (std) values of +3.1 (2.5) and +0.8 (0.8) W m$^{-2}$ at BOA and TOA, respectively (see Table 5)."

**Reviewer's comment (R2C19):** Figure 1(b): Suggest adding fitting statistics to the plot.

**Authors' response:** Following the reviewer's suggestion, the fitting statistics has been added to Figures 1b and 1c, and also in Figure S3 of the Supplementary Material.

[Figure]

**Figure 1. (a)** Hourly land surface temperature (LST, in °C), where the red dots represent the cases coincident with lidar measurements; AERONET geometric median radius ($r_g$, in μm ) and standard deviation ($\sigma_g$) for the **(b)** fine and **(c)** coarse modes, where the dashed lines represent the linear fitting of $r_g$ over time; Episode-averaged values of **(d)** the Mie-derived normalized spectral extinction ($\alpha_{LW}\ (Mie)/\alpha_{532}(Mie)$) (see Eq. 3), **(e)** asymmetry factor ($g_{LW}$), and **(f)** single scattering albedo ($\omega_{LW}$), for the fine (blue), coarse (red) and total (yellow) modes. All the panels refer to El Arenosillo/Huelva (ARN) station; for the rest of stations, see the Supplementary Material.

[Figure]

**Figure S3.** Hourly geometric median radius ($r_g$, µm; in black), and standard deviation ($\sigma_g$, µm; in blue), as derived from AERONET data (see Eq. 1 in the manuscript), at the five lidar stations (from NE to SW, by decreasing latitude): Barcelona (BCN), Torrejón/Madrid (TRJ), Évora (EVO), Granada (GRA) and El Arenosillo/Huelva (ARN) for: a) the fine mode, and b) the coarse mode. The dashed lines represent the linear fitting along the period. The slope of each linear fitting ($\gamma$) can be found in Table 2 in the manuscript.

**Reviewer's comment (R2C20):** Figure 2: Add indicators for days with daily DD DOD532 greater or less than 0.5, since this threshold is referenced multiple times. Explain the inset numbers in panels (a) and (b).

**Authors' response:** To preserve the clarity of Figure 2, authors have included an additional figure in the Supplementary Material (Figure S4) to show the temporal evolution of DOD$^{532}$ through the episode, highlighting the days exhibiting hourly DOD$^{532}$ > 0.5.

[Figure]

**Figure S4.** Temporal evolution of the total dust optical depth at 532 nm (DOD$^{532}$) over the five Iberian lidar stations as latitude decreases (from up to down panels): a) BCN, b) TRJ, c) EVO, d) GRA, and e) ARN. The green bars corresponds to the profiles used in the LW simulations. The dashed line corresponds to DOD$^{532}$=0.5. Days with hourly DOD$^{532}$ > 0.5 are marked in shadow.

Moreover, the following text has been added to specially introduce the specific days:

Page 7, lines 280-283: "A detailed description of the dust incidence of the Saharan intrusion by crossing the Iberian Peninsula is provided in López-Cayuela et al. (2023). In addition, the temporal evolution of the DOD[532] for the five lidar stations is shown in Figure S4 of the Supplementary Material, where the particular days with high aerosol loads (i.e., hourly DOD[532] > 0.5) are also indicated, occurring mainly between 27 March and 1 April 2021 at several stations. "

**Reviewer's comment (R2C21):** Table 2: Suggest adding standard errors to the fitted slope values.

**Authors' response:** Following the reviewer's comment, the standard error has been added to the fitted slope ($\gamma(r_g)$) values in Table 2.

**Table 2. Episode-averaged median radius ($r_g$, μm) and standard deviation ($\sigma_g$, μm) at the five lidar stations: Barcelona (BCN), Torrejón/Madrid (TRJ), Évora (EVO), Granada (GRA) and El Arenosillo/Huelva (ARN) for the fine and coarse modes. The slope of each linear fitting (γ, % μm day$^{-1}$) and its standard error (in brackets) is also shown.**

|  |  | ARN | GRA | EVO | TRJ | BCN |
|---|---|---|---|---|---|---|
| Fine mode | $r_g$ | +0.076 | +0.093 | +0.083 | +0.067 | +0.059 |
|  | $\gamma(r_g)$ | -0.42 (0.06) | +0.57 (0.18) | -0.38 (0.07) | +0.44 (0.05) | +0.75 (0.12) |
|  | $\sigma_g$ | +0.613 | +0.651 | +0.624 | +0.575 | +0.552 |
| Coarse mode | $r_g$ | +0.471 | +0.584 | +0.529 | +0.878 | +0.578 |
|  | $\gamma(r_g)$ | -0.36 (0.39) | -1.96 (1.55) | -0.59 (0.36) | +2.00 (0.77) | +6.90 (0.96) |
|  | $\sigma_g$ | +0.585 | +0.584 | +0.592 | +0.653 | +0.642 |